theoretical biology/synthetic biology/statistics

dynamical systems, genetic vector control, *Plasmodium falciparum*, next generation matrix, transient dynamics, *Wuchereria bancrofti*

**Author for correspondence:**
Geoffrey R. Hosack
e-mail: geoff.hosack@csiro.au

# Quantifying the risk of vector-borne disease transmission attributable to genetically modified vectors

## Geoffrey R. Hosack, Adrien Ickowicz and Keith R. Hayes

Commonwealth Scientific and Industrial Research Organisation, Data61, Hobart, Tasmania, Australia

   GRH, 0000-0002-6462-6817; AI, 0000-0001-7861-7340; KRH, 0000-0003-4094-3575

The relative risk of disease transmission caused by the potential release of transgenic vectors, such as through sterile insect technique or gene drive systems, is assessed with comparison with wild-type vectors. The probabilistic risk framework is demonstrated with an assessment of the relative risk of lymphatic filariasis, malaria and o'nyong'nyong arbovirus transmission by mosquito vectors to human hosts given a released transgenic strain of *Anopheles coluzzii* carrying a dominant sterile male gene construct. Harm is quantified by a logarithmic loss function that depends on the causal risk ratio, which is a quotient of basic reproduction numbers derived from mathematical models of disease transmission. The basic reproduction numbers are predicted to depend on the number of generations in an insectary colony and the number of backcrosses between the transgenic and wild-type lineages. Analogous causal risk ratios for short-term exposure to a single cohort release are also derived. These causal risk ratios were parametrized by probabilistic elicitations, and updated with experimental data for adult vector mortality. For the wild-type, high numbers of insectary generations were predicted to reduce the number of infectious human cases compared with uncolonized wild-type. Transgenic strains were predicted to produce fewer infectious cases compared with the uncolonized wild-type.

## 1. Introduction

### 1.1. Genetic vector control and risk of disease transmission

The relative risk of disease transmission is investigated for the release of a transgenic vector as part of an entomological study. The study was conducted as part of a staged research

development pathway that investigates the potential for gene drive technology to suppress malaria vectors and thereby reduce malaria transmission [1]. For gene drive technology applied to vector control of mosquitoes, probabilistic safety assessments should compare the risks of disease transmission by transgenic mosquitoes relative to wild-type mosquitoes of the same genetic background [2,3]. The theoretical framework developed here is broadly applicable and may be used to support general assessments for the relative risk of disease transmission by transgenic and wild-type vectors. The risk approach is demonstrated for transgenic vectors in a genetically engineered sterile insect technique (SIT) entomological study.

Vector control is a proven means to allay human morbidity and mortality, and increasingly genetic engineering (GE) is proposed as an effective way to control mosquito vectors [4–6]. SIT is a vector control technique that may benefit from genetic production methods [7]. Conventional SIT uses chemical or irradiative methods to sterilize male insects and has been successfully applied to insect pest and vector populations [8]. Genetically engineered SIT could be used to increase the effectiveness of SIT deployed to control populations of malaria vectors in Africa [9]. In 2019, an estimated 229 million cases of malaria caused 409 000 deaths of which 94% were located in Africa, mostly among children under 5 years of age [10]. The mosquito species group *Anopheles gambiae* sensu lato (s.l.) is a dominant malaria vector in Africa [11] that also vectors neglected tropical diseases such as lymphatic filariasis, which is currently subject to an ongoing global elimination programme [12], and potentially emergent diseases such as o'nyong'nyong [13].

In *A. gambiae* s.l., self-limiting dominant sterile males, a form of genetically engineered SIT, have been developed using homing endonuclease genes (HEGs) in a phased development pathway that can be extended to a self-sustaining population suppression gene drive system [14–16]. The insectary population with the genetically engineered sterile male construct was derived by transgenesis of a laboratory colony strain known as G3 [17]. The strain used for release in the entomological study began by backcrossing insectary-colonized local *A. coluzzii* wild-type mosquitoes with the transgenic laboratory lineage. Backcrosses between the transgenic lineage and local wild-type target population within the insectary production facility can improve genetic compatibility and efficacy of the release in SIT programmes and field studies [7]. The backcrosses between the wild-type strain and transgenic (GE) strain allow introgression of the wild-type genetic background into the GE strain. The genetic composition of the colonized wild-type population within the insectary may diverge from the local field wild-type population with each generation in the insectary [18,19] with implications for genetic vector management strategies [3,20]. Therefore, the risk assessment approach is generically demonstrated on the accidental release of insectary-colonized wild-type vectors in addition to the genetically engineered strain.

## 1.2. Risk assessment endpoint

In the case of developing transgenic vector technology, risk assessments should investigate the risk of adverse effects in the vector including adverse changes to vectorial capacity and vector competence [21]. The staged development pathway for gene drive systems should include insectary studies, contained trials and semi-field studies, among other factors, and also consideration of self-limiting precursors such as SIT [2]. The staged development pathway of transgenic mosquitoes requires evaluation of the risk posed by transgenic mosquitoes relative to unmodified mosquitoes [22], but the translation from laboratory-based outcomes to field experiments introduces uncertainty that should be assessed. The use of homing endonucleases in the transgenic sterile male release examined in this study could be extended to gene drive applications [14–16]. However, the current scientific advice is that gene drive technology requires further scientific research and risk assessment before any field trials may be considered [23]. Risk assessments that estimate the harm on human health imposed by transgenic versus wild-type vectors are required before a gene drive candidate may advance to field trials [3], and such risk assessments should be probabilistic [2,23].

The relative risk of increased disease transmission from the release of insectary-bred transgenic vectors versus wild-type is an assessment endpoint of primary interest [21]. Female mosquitoes can transmit vector-borne diseases by feeding on humans, and so the accidental release of female mosquitoes is an important consideration for risk assessment of all SIT programmes and field studies, not just GE-based methods [8,24]. For the demonstrated risk assessment, the comparison of interest is between the transmission properties of insectary-colonized female mosquitoes and the local uncolonized wild-type strain for the following example pathogens vectored by anopheline mosquitoes in the Sudanian zone of Burkina Faso and the neighbouring Sudano-Guinean area of West Africa: (i) the neglected tropical

disease lymphatic filariasis caused by the nematode *Wuchereria bancrofti* [25,26]; (ii) malaria (*Plasmodium falciparum*) for which *Anopheles gambiae* s.l. is a dominant vector in Africa [11]; and (iii) the periodically re-emerging o'nyong'nyong alphavirus (ONNV) that uniquely among alphaviruses is primarily vectored by *Anopheles* mosquitoes [13]. The example pathogens comprise a multicellular animal (*W. bancrofti*), a unicellular protist eukaryote (*P. falciparum*), and an arbovirus (ONNV).

The scientific risk assessment uses a decision analytic framework that incorporates both probability and loss [27] to assess the relative risk of disease transmission given an accidental release of insectary-colonized wild-type or genetically engineered strains versus uncolonized wild-type female mosquitoes. The scientific risk assessment supports evidence-based decision-making under uncertainty by evaluation of the expected losses that arise from alternative decisions while coherently accounting for expert domain knowledge and empirical data [27,28]. Expected loss codifies the definition of risk in terms of the probability and severity of harm, which has been identified as best practice for evaluation of transgenic technologies such as SIT and gene drive applications [2,21–23]. The approach thereby provides a coherent calculus for considering uncertainty in light of different forms of evidence including theory and experimental or observational data to advise evidence-based decision-making.

# 2. Biomathematical model

## 2.1. Disease transmission indices

The standard metric for assessment of disease transmission in mathematical models is the basic reproduction number, denoted by $\mathcal{R}_0$, which is defined as the expected number of secondary cases that arise from a single infected case, or infectious case, in a fully susceptible population [29]. This metric is also a threshold index, such that if $\mathcal{R}_0 < 1$ then the disease-free equilibrium (DFE, defined by fully susceptible populations with no infectious individuals) is locally stable, so that disease transmission is not sustained in the long term following the introduction of a small number of infectious individuals. Let the basic reproduction number $\mathcal{R}_0^{j(G,B)}$ apply to the transmission of pathogen $j$ given a release of either transgenic or wild-type colonized vectors in an entomological study. The released strain is either the genetically engineered vector with at least one insectary generation ($G > 0$) and at least one backcross $B$ ($0 < B \leq G$) or the wild-type strain with no backcrosses ($G \geq 0$, $B = 0$). For example, the basic reproduction number for the uncolonized wild-type vector with no insectary generations and no backcrosses is $\mathcal{R}_0^{j(G=0,B=0)}$. The index $j \in \{l, m, o\}$ is evaluated for the infectious pathogens of lymphatic filariasis (*W. bancrofti*), malaria (*P. falciparum*) and ONNV, respectively.

It remains to define the basic reproduction numbers $\mathcal{R}_0^{j(G,B)}$ such that these indices are theoretically observable by either experimental or observational studies. Then the above risk assessment endpoint becomes a measurable assessment endpoint and thus a valid scientific target for a probabilistic risk assessment [30]. In particular, for the basic reproduction number, different formulations of the underlying dynamic model of disease transmission can change the relative importance of parameters such as the larval emergence rate [31,32]. The model scenario seeks to distil the comparison among strains with an emphasis on the vector-based parameters of disease transmission that contribute to the basic reproduction number (e.g. vectorial capacity, [33]). The model scenario therefore adopts many of the base assumptions used by the so-called Ross–Macdonald theory [34] that builds on the original mathematical model proposed by [35] and the heuristic model of [36]. The transmission dynamics are modelled by a dynamic system of susceptible and infectious states for both hosts and vectors (§2.3). Additionally, incubation periods of the pathogen in the host (intrinsic) and vector (extrinsic) introduce delays in pathogen transmission that are explicitly incorporated into the dynamic model.

The basic reproduction number $\mathcal{R}_0^{j(G,B)}$ is a long-term average of pathogen transmission that applies if the released strain persists indefinitely (§2.4). A corresponding thought experiment is one where the released cohort successfully invades an empty or vacated niche and increases to carrying capacity as it reproduces at the site of release. Such a scenario would in actuality be difficult to achieve for a transgenic vector population carrying a sterile male construct because the strong fitness cost of the transgene implies its rapid loss from the established population in the absence of subsequent releases [37]. The transgenic vector would be unlikely to persist at carrying capacity in the long term. The actual exposure to the transgenic vector is likely to be less than what is assumed by an estimate based on the basic reproduction number that assumes a persisting transgenic vector population. Nevertheless, as a theoretical construct $\mathcal{R}_0$ usefully directs focus to key parameters that determine the transmission potential for vector-borne diseases.

The basic reproduction number $\mathcal{R}_0^{j(G,B)}$ as derived from the so-called next generation matrix [38,39] is shown to have a clear biological interpretation (§2.5). The next generation matrix is then used to derive the expected number of secondary infectious cases caused by a single released cohort (§2.6). The derived index of transmission for a single release is denoted by $\mathcal{R}_s^{j(G,B)}$. It describes the transitory effect of a single release, and is shown by the next generation matrix to provide an upper bound for transmission given additional assumptions on the infected status of released vectors as summarized by the index $\mathcal{R}_{s'}^{j(G,B)}$ (§2.7). The indices $\mathcal{R}_s^{j(G,B)}$ and $\mathcal{R}_{s'}^{j(G,B)}$ both focus on the transitory effects of a single released cohort into an empty or vacated niche. These indices complement the long-term theoretical measure of disease transmission described by the basic reproduction number. The three disease transmission indices enable comparison of predicted harm between genetically engineered and conventional strains [2,21,22] while controlling for other factors attributable to insectary rearing and genetic background.

## 2.2. Evidence-based decision-making under uncertainty

The logarithm function is a common loss or utility (negative of loss) function used in decision theory [28]. An interpretation of logarithmic utility applied to social welfare decisions is given by [40]. The net logarithmic loss of releasing an insectary-colonized strain relative to uncolonized wild-type is given by the quantity $\log_{10}(\mathcal{R}_0^{j(G>0,B\geq0)}/\mathcal{R}_0^{j(G=0,B=0)})$, where the transmission rate parameters that define the basic reproduction numbers are uncertain. Analogously, the quantity $\log_{10}(\mathcal{R}_s^{j(G>0,B\geq0)}/\mathcal{R}_0^{j(G=0,B=0)})$ is the net loss of the transient impact of the single released cohort defined by $\mathcal{R}_s^{j(G>0,B\geq0)}$ relative to the background transmission of an established uncolonized wild-type population.

The net logarithmic losses have the following interpretation. A positive (negative) net logarithmic loss of $M$ means that the number of secondary infectious cases for the insectary-colonized strain is $M$ orders of magnitude greater (lower) than for the local uncolonized wild-type. Moreover, the net logarithmic loss explicitly depends on a measure of attributable risk. The units of the basic reproduction number are dimensionless but interpretable as 'cases/case' [38]; that is, it is the number of secondary cases arising from the primary case. The causal risk ratio ([41], Ch. 3) of secondary cases that arise from exposure to insectary-colonized vectors ($G>0$), relative to uncolonized wild-type, is then the ratio $\mathcal{R}_0^{j(G>0,B\geq0)}/\mathcal{R}_0^{j(G=0,B=0)}$ of basic reproduction numbers. In a closed population, this risk ratio is a measure of attributable risk to exposure from a released strain of insectary-colonized mosquitoes relative to uncolonized wild-type.

The causal risk ratio is an example of a counterfactual or potential outcome definition for effect ([41], Ch. 3): the causal risk ratio compares disease transmission risk under two different hypothetical scenarios of which one but not the other may actually occur. The basic reproduction number is a long-term measure of transmission risk (§2.5), and so the alternative indices of transmission risk given by $\mathcal{R}_s^{j(G,B)}$ and $\mathcal{R}_{s'}^{j(G,B)}$ are developed for hypothetical scenarios that consider the transitory effect of a released cohort of insectary-colonized vectors (§2.6). The corresponding transitory and long-term causal risk ratios are derived from the dynamic system described in the following section.

## 2.3. System of delay differential equations

A system of delay differential equations that described the transmission of malaria between host and vector populations of constant size while accounting for both intrinsic and extrinsic incubation periods was proposed by [42]. Here, this model is generalized to allow (i) a more ecologically realistic non-constant vector population that exhibits density-dependent growth following release, (ii) the vector mortality rate to vary by infection status, and (iii) the human recovery rate to change when an infected host becomes infectious. Let $k$ denote a strain with $G$ insectary generations and $B$ backcrosses. For each pathogen $j$ and release scenario of strain $k$, the model is given by

$$\frac{dx(t)}{dt} = \frac{a_k b_{jk}}{H} z_k(t-\omega_j)[H-x(t-\omega_j)]\,e^{-\kappa_j\omega_j} - r_j x(t) \tag{2.1a}$$

$$\frac{dy_k(t)}{dt} = (\lambda_k - \mu_k)\,y_k(t)\left[\frac{v-y_k(t)}{v}\right] - \frac{a_k c_{jk}}{H}\,y_k(t)\,x(t), \quad \lambda_k > \mu_k \tag{2.1b}$$

and

$$\frac{dz_k(t)}{dt} = \frac{a_k c_{jk}}{H}\,y_k(t-\tau_{jk})\,x(t-\tau_{jk})\,e^{-\mu_{jk}\tau_{jk}} - \mu_{jk}\,z_k(t). \tag{2.1c}$$

Equation (2.1a) gives the change in the number $x(t)$ of infectious human hosts at time $t$. Human hosts, where $H$ is the total human population, become infected by contact with infectious vectors at time $t-\omega_j$. The number of bites on humans per day made by a vector is given by $a_k$ and the number of

**Table 1.** Parameters appearing in the dynamic model given by equation (2.1). The elicited parameters vary by disease $j \in \{l, m, o\}$ (lymphatic filariasis, $l$; malaria, $m$; and ONNV, $o$) and strain $k$ of released vectors (wild-type or genetically engineered). The relative risk approach decreased elicitation load and allowed experts to focus on entomological transmission parameters. Elicited parameters accounted for variation in the genetic background of mosquito vectors as described by number of insectary generations and backcrosses between wild-type and genetically engineered vectors.

| parameter | elicited | definition |
|---|---|---|
| $a_k$ | yes | number of bites on human per day per mosquito |
| $b_{jk}$ | yes | transmission efficiency from female mosquitoes to humans |
| $c_{jk}$ | yes | transmission efficiency from humans to female mosquitoes |
| $\mu_k$ | yes | mortality rate of uninfected female mosquitoes |
| $\mu_{jk}$ | yes | mortality rate of infected female mosquitoes |
| $\tau_{jk}$ | yes | duration of extrinsic incubation period |
| $\lambda_k$ | no | larval emergence rate |
| $v$ | no | environmental carrying capacity of female mosquitoes |
| $v_0$ | no | number of released female mosquitoes |
| $H$ | no | number of human hosts |
| $\omega_j$ | no | duration of intrinsic incubation period |
| $\kappa_j$ | no | host recovery rate during intrinsic incubation period |
| $r_j$ | no | host recovery rate after intrinsic incubation period |

bites on humans per human is $a_k/H$. Of the bites made by infectious vectors on humans, a proportion $b_{jk}$ transmits the pathogen to the human host. Infected humans recover at rate $\kappa_j$ during the intrinsic incubation period of duration $\omega_j$. At the end of the incubation period, a proportion of human hosts $e^{-\kappa_j \omega_j}$ have recovered and are again susceptible. Hosts not recovered at the end of the intrinsic incubation period become infectious and recover at rate $r_j$.

Equation (2.1$b$) gives the change in the number of uninfected vectors, $y_k(t)$. The vector population exhibits logistic growth with carrying capacity $v$. The intrinsic growth parameter, which is assumed positive to avoid pathological behaviour, is defined by the difference between the larval emergence rate $\lambda_k$ and the uninfected vector mortality rate $\mu_k$. Uninfected vectors become infected with probability $c_{jk}$ after biting an infectious human and are lost from the uninfected vector compartment.

Equation (2.1$c$) gives the change in the number of infectious vectors, $z_k(t)$. Vectors become infected by contact with infectious human hosts and decay at mortality rate $\mu_{jk}$. After the end of the extrinsic incubation period $\tau_{jk}$, a proportion $e^{-\mu_{jk} \tau_{jk}}$ survive. The parameters of equation (2.1) are summarized in table 1.

## 2.4. Threshold index for globally stable disease-free equilibrium

Equilibria associated with equation (2.1) can be obtained by ignoring time delays, setting the system of ordinary differential equations (ODEs) to zero, and then solving for each of the variables. The DFE is defined by the point $(x, y, z) = (0, v, 0)$, where there are no infectious vectors or humans and the number of uninfected vectors is at the environmental carrying capacity $v$. It is shown below that the DFE is globally asymptotically stable if

$$\mathcal{R}_0^{jk} = \frac{a_k^2 b_{jk} c_{jk} m \, e^{-\kappa_j \omega_j} \, e^{-\mu_{jk} \tau_{jk}}}{r_j \mu_{jk}} < 1, \tag{2.2}$$

where $m = v/H$ is the ratio of the vector carrying capacity to the human host population. This expression establishes $\mathcal{R}_0^{jk}$ as a threshold index that determines whether or not the pathogen can become established in a susceptible population. The above expression for $\mathcal{R}_0^{jk}$ is similar to the expression derived by [42], generalized here such that the vector population size may vary, the vector mortality rate may depend on infection status, and the human recovery rate may depend on the intrinsic incubation period. In the following, the next generation matrix [38,39] is used to derive $\mathcal{R}_0^{jk}$ (an alternative derivation is

used in [42]) and thereby inform its biological interpretation for long-term disease transmission (§2.5). However, the transgenic strain is unlikely to persist in the long term, contrary to the assumptions of the basic reproduction number (§2.1). The next generation matrix and its biological interpretation is therefore applied to develop transitory indices of disease transmission for single cohorts of released vectors, $\mathcal{R}_s^{jk}$ and $\mathcal{R}_{s'}^{jk}$ (see §§2.6 and 2.7) that allow comparison between wild-type and transgenic strains.

For all initial data that satisfies equation (2.1) with a non-zero number of uninfected vectors, it is evident from equation (2.1) that the number of uninfected vectors asymptomatically approaches the carrying capacity, $y_k \to v$, if there are no infectious hosts or vectors. The DFE of equation (2.1) is thus asymptotically attained if it can be shown that $(x(t), z_k(t)) \to (0, 0)$ as $t \to \infty$. Let the initial data satisfy equation (2.1) such that the number of initial uninfected vectors is non-zero and does not exceed the carrying capacity $v$. With these initial data, equations (2.1$a$) and (2.1$c$) are bounded above by

$$\frac{dx(t)}{dt} \le a_k b_{jk} z_k(t - \omega_j) e^{-\kappa_j \omega_j} - r_j x(t) \tag{2.3$a$}$$

and

$$\frac{dz_k(t)}{dt} \le (a_k/H) c_{jk} v x(t - \tau_{jk}) e^{-\mu_{jk} \tau_{jk}} - \mu_{jk} z_k(t). \tag{2.3$b$}$$

The right-hand side of equation (2.3$a$) increases with the delayed variable $z_k(t - \omega_j)$ and decreases with the non-delayed variable $x(t)$. The right-hand side of equation (2.3$b$) increases with the delayed variable $x(t - \tau_{jk})$ and decreases with the non-delayed variable $z_k(t)$. The configuration of the sign pattern thus obtained shows that the linear subsystem with inequalities set to equalities in equation (2.3) is cooperative and irreducible ([43], Ch. 5). This subsystem therefore possesses a quasi-monotone condition ([43], Ch. 5) such that its solution with the same initial data as in equation (2.1) bounds $x(t)$ and $z_k(t)$.

Setting the delays to zero in the linear system given by equation (2.3), where the inequalities therein are replaced with equalities, produces the associated ODE system

$$\frac{dx}{dt} = a_k b_{jk} z_k(t) e^{-\kappa_j \omega_j} - r_j x(t) \tag{2.4$a$}$$

and

$$\frac{dz_k}{dt} = (a_k/H) c_{jk} v x(t) e^{-\mu_{jk} \tau_{jk}} - \mu_{jk} z_k(t). \tag{2.4$b$}$$

By corollary 5.5.2 of [43] that applies to cooperative and irreducible systems, the linear stability analysis applied to equation (2.4) at the equilibrium $(x, z_k) = (0, 0)$ also applies to the linear system defined by replacing the inequalities with equalities in equation (2.3). If this equilibrium is stable in equation (2.4) then $(x(t), z_k(t)) \to (0, 0)$ as $t \to \infty$, and so also in equation (2.3) with inequalities replaced by equalities, and hence in equation (2.1) as well. The DFE is therefore globally asymptotically stable if the equilibrium $(x, z_k) = (0, 0)$ of equation (2.4) is stable.

The next generation matrix of linear stability analysis [38] was applied by [44] to variants of the Ross–Macdonald ODE model similar to equation (2.4). This method informs the biological interpretation below (§2.5) and is now illustrated for equation (2.4) with equilibrium $(x, z_k) = (0, 0)$. The Jacobian matrix with entries composed of the partial derivatives of equation (2.4) is

$$J = \begin{bmatrix} -r_j & a_k b_{jk} e^{-\kappa_j \omega_j} \\ a_k c_{jk}(v/H) e^{-\mu_{jk} \tau_{jk}} & -\mu_{jk} \end{bmatrix}, \tag{2.5}$$

where the dependence of $J$ on the choice of disease $j$ and strain of vector $k$ is suppressed to simplify notation. Separating the off-diagonal entries into the non-negative infection matrix $F$ and the negative of the diagonal entries into the diagonal matrix $V$ obtains the so-called next generation matrix $FV^{-1}$ [39]. Theorem 2 of [39] then applies, which shows that

$$\rho(FV^{-1}) < 1 \Leftrightarrow s(J) < 0,$$

where $\rho(A)$ is the spectral radius of the matrix $A$ and $s(A)$ is its spectral abscissa. It can be shown that $\rho(FV^{-1}) = \sqrt{\mathcal{R}_0^{jk}}$. The equilibrium $(x, z_k) = (0, 0)$ of equation (2.4) is then stable if $\mathcal{R}_0^{jk} < 1$. Therefore, the condition of equation (2.2) implies global asymptotic stability of the DFE.

## 2.5. Interpretation of basic reproduction number

The quantity of interest is the expected number of secondarily infectious hosts produced by a single infectious host, when the host and vector populations are otherwise fully susceptible (*sensu* [45]). The system defined by equation (2.3) approximates this scenario: equations (2.1*a*) and (2.1*c*) are well approximated by the bounds of equations (2.3*a*) and (2.3*b*) for small perturbations near the DFE. These bounds are attained with equality, for example, with perturbations of one infectious host from the DFE such that equation (2.1) is satisfied with the conditions $y_k(t - \tau_{jk}) = v$, $x(t - \tau_{jk}) = 1$ and $x(t - \omega_j) = 0$, while allowing $x(t) \geq 0$ with infectious vectors $z_k(t - \omega_j) \geq 0$ and $z_k(t) \geq 0$. The long-term behaviour of the system defined by equation (2.3) is identified with the ODE system $d\phi(t)/dt = (F - V)\phi(t)$ with $\phi(t) = [x(t), z_k(t)]^\top$ from equation (2.4). In §2.4, the next generation matrix is given by,

$$FV^{-1} = \begin{bmatrix} 0 & a_k b_{jk} e^{-\kappa_j \omega_j}/\mu_{jk} \\ a_k c_{jk}(v/H)e^{-\mu_{jk}\tau_{jk}}/r_j & 0 \end{bmatrix}.$$

Each entry in the next generation matrix provides the expected number of infectious secondary cases produced by a primary infectious case in a single transmission generation. For example, the expected number of secondary infectious vectors produced by a host while infectious is given by $a_k c_{jk}(v/H) e^{-\mu_{jk}\tau_{jk}}/r_j$. An infectious vector is expected to produce $a_k b_{jk} e^{-\kappa_j \omega_j}/\mu_{jk}$ secondary infectious hosts in a single transmission generation.

Given the initial condition of a single infectious host, $\phi(0) = [1, 0]^\top$, the expected number of secondary infectious vectors in one transmission generation is obtained from the next generation matrix as $FV^{-1}\phi(0) = \int_0^\infty F e^{-Vt}\phi(0)\,dt$ [46]. The number of infectious hosts and vectors after $\gamma$ transmission generations is given by $(FV^{-1})^\gamma\phi(0)$ [38]. For human and mosquito disease transmission, two transmission generations are required to complete a full cycle from an infectious host to an infectious vector and back [38,45,46], so that $\gamma = 2$. The result $\mathcal{R}_0^{jk} = [(FV^{-1})^2\phi(0)]_1$ follows, where $[\alpha]_i$ is the $i$th entry of the vector $\alpha$ and $\mathcal{R}_0^{jk}$ is equal to the expression derived in §2.4 (equation (2.2)).

The basic reproduction number $\mathcal{R}_0^{jk}$ therefore has the desired interpretation as the average number of secondary infectious hosts for disease $j$ that arise from a single typical infectious host in an otherwise susceptible human population given a released vector strain $k$ that becomes established when introduced into an empty or vacated niche at the release site. However, the transgenic strain is unlikely to persist in the long term (§2.1). In the below §§2.6 and 2.7, the next generation matrix is therefore used to derive transitory indices that focus on disease transmission for a single cohort introduced into an empty or vacated niche at the release site.

## 2.6. Exposure to released cohort

Another important point of comparison is the risk of a transient impact of a released insectary vector on disease transmission compared with local background levels of transmission [21]. For a release scenario focused on a single cohort of $v_0$ released vectors, an experiment could in principle be performed by larviciding after release such that the larval emergence rate is zero, so that $\lambda_k = 0$ in equation (2.1). The trivial equilibrium is then globally stable and the basic reproduction number $\mathcal{R}_0^{jk}$ is no longer useful as a threshold parameter for stability of the DFE. The size of the single cohort cannot increase in this scenario and equation (2.1*b*) is modified to

$$\frac{dy_k(t)}{dt} = -\mu_k y_k(t) - (a_k/H)c_{jk}y_k(t)x(t).$$

The maximum number of infectious vectors possible is then $v_0$. The bound for infectious vectors previously given by equation (2.3*b*) to derive $\mathcal{R}_0^{jk}$ (§2.4) may therefore be adjusted for the single cohort scenario with the substitution of $v_0$ for the carrying capacity $v$. Note that this adjusted bound with $v_0$ substituted for $v$ in equation (2.3*b*) holds whether or not the number of released vectors $v_0$ exceeds the carrying capacity $v$. The infection rate matrix $F$ is thereby modified to $F_s$, where the parameter $v$ in $F$ is substituted for by $v_0$ in $F_s$. The logic applied to the biological interpretation (§2.5) proceeds otherwise unchanged to develop an expected transmission index for the single release given by,

$$\mathcal{R}_s^{jk} = [(F_s V^{-1})^2\phi(0)]_1 = \frac{a_k^2 b_{jk} c_{jk}(v_0/H)e^{-\kappa_j\omega_j}e^{-\mu_{jk}\tau_{jk}}}{r_j\mu_{jk}}. \tag{2.6}$$

Three causal risk ratios are now designed to assess the counterfactual or potential outcomes given alternative hypothetical scenarios (§2.2). The causal risk ratios constructed from either $\mathcal{R}_0^{jk}$ or $\mathcal{R}_s^{jk}$,

given by equations (2.2) and (2.6), are equivalent for a colonized strain $k$ with $G$ insectary generations and $B$ backcrosses compared with uncolonized wild-type because $\mathcal{R}_0^{j(G>0,B\geq0)}/\mathcal{R}_0^{j(G=0,B=0)} = \mathcal{R}_s^{j(G>0,B\geq0)}/\mathcal{R}_s^{j(G=0,B=0)}$. For either choice of index, the parameters $H$, $\omega_j$, $\kappa_j$ and $r_j$ along with $v$ or $v_0$ (table 1) cancel from the causal risk ratio. The causal risk ratios and net logarithmic losses (§2.2) are equivalent for the two choices of comparison by either the long-term index $\mathcal{R}_0^{jk}$ or instead the transitory index $\mathcal{R}_s^{jk}$.

It is also possible to develop a third causal risk ratio composed of both $\mathcal{R}_0^{jk}$ and $\mathcal{R}_s^{jk}$. For example, the causal risk ratio $\mathcal{R}_s^{j(G>0,B\geq0)}/\mathcal{R}_0^{j(G=0,B=0)}$ compares the transitory disease transmission risk from a released cohort of colonized vectors for the scenario considered in this section to the long-term background transmission from a persisting local wild-type population of uncolonized vectors. As before many parameters cancel out; however, in this comparison the carrying capacity $v$ and size of initial release $v_0$ no longer cancel out of the causal risk ratio.

## 2.7. Exposure to release of uninfected vectors

Consider a release for an entomological study that results in the accidental release of $v_0$ female vectors that are uninfected for the pathogen $j$. At the time of release $t = 0$, a maximum of $v_0$ vectors can become infected. Of the infected vectors, a proportion $e^{-\tau_{jk}\mu_{jk}}$ survive at the end of the extrinsic incubation period. The bound for infectious vectors previously given by equation (2.3b) may therefore be altered for this scenario with the substitution for $v$ by the quantity $v_0 e^{-\tau_{jk}\mu_{jk}}$: the infection rate matrix $F$ is modified to $F_{s'}$, where now the parameter $v$ in $F$ is substituted for by $v_0 e^{-\tau_{jk}\mu_{jk}}$ to obtain $F_{s'}$. After two transmission generations (see §2.5), the expected number of infectious hosts is given by $\mathcal{R}_{s'}^{jk} = [(F_{s'}V^{-1})^2\phi(0)]_1$. From equation (2.6), there is then the bound given by

$$\mathcal{R}_{s'}^{jk} = \frac{a_k^2 b_{jk} c_{jk}(v_0 e^{-\mu_{jk}\tau_{jk}}/H) \, e^{-\kappa_j\omega_j} \, e^{-\mu_{jk}\tau_{jk}}}{r_j\mu_{jk}} = e^{-\mu_{jk}\tau_{jk}}\mathcal{R}_s^{jk} < \mathcal{R}_s^{jk}. \tag{2.7}$$

The quantity $\mathcal{R}_{s'}^{jk}$ is therefore an upper bound on the average number of secondary infectious cases that arise due to the single released cohort of $v_0$ uninfected vectors given a single infectious host. The index $\mathcal{R}_{s'}^{jk}$ exponentially decreases from the upper bound $\mathcal{R}_s^{jk}$ with the infected mortality rate $\mu_{jk}$ and duration of the extrinsic incubation period $\tau_{jk}$.

# 3. Statistical model

## 3.1. Generalized linear model

The parameters in table 1 for equation (2.1) were defined as average daily rates, averaged over a year, for non-aestivating mosquitoes. The parameters were considered to independently describe the various biological aspects of transmission. Let $\psi$ denote one of the elicited independent parameters of table 1. For each such $\psi$, a generalized linear model [47] was developed,

$$\zeta \sim p(\psi) \tag{3.1a}$$
$$\psi = h^{-1}(\eta) \tag{3.1b}$$
$$\eta = X\beta \tag{3.1c}$$
$$\beta \sim N(\delta, \Sigma). \tag{3.1d}$$

In equation (3.1a), the $d$-dimensional vector of observations $\zeta$ of a vectorial competence parameter selected from table 1 is generated by a natural exponential family model $p(\psi)$ [48]. The $d$-dimensional mean vector $\psi$ is related to the linear predictor $\eta$ by a monotonic link function $h(\cdot)$ in equation (3.1b).

In equation (3.1c), the $d \times p$ matrix $X$ captures the contribution of different strains. The insectary population with the genetically engineered sterile male construct, denoted by Ag(DSM)2 (for *A. gambiae* dominant sterile male, line 2), was derived by transgenesis of a laboratory colony strain known as G3 [17]. The strain used for release in the entomological study is denoted by Ac(DSM)2 because it began by backcrossing insectary-colonized local *A. coluzzii* wild-type mosquitoes with the Ag(DSM)2 lineage. The nomenclature used to identify these strains is summarized in table 2. The matrix $X$ is derived from the predictors $u_j$ that are inputs into the model (table 3, see equation (3.2) below). The $p \times 1$ dimensional vector of unknown linear coefficients $\beta$ is assigned a multivariate normal distribution with mean vector $\delta$ and covariance matrix $\Sigma$ in equation (3.1d).

The human feeding rate and the extrinsic incubation period parameters have positive support, $(0, \infty)$, and used the log link, $\eta = h(\psi) = \log\psi$. All other parameters (transmission efficiencies and the probability

**Table 2.** Nomenclature for genetic strains.

| label | description |
| --- | --- |
| G3 | original laboratory strain |
| Ag(DSM)2 | G3 laboratory strain with GE sterile male construct |
| Ac(WT) | local *A. coluzzii* wild-type |
| Ac(DSM)2 | Ag(DSM)2 with Ac(WT) backcrosses |

**Table 3.** Covariates used to construct the design matrices for the generalized linear models.

| covariate | description |
| --- | --- |
| $\xi_G$ | number of Ac(WT) generations in insectary colony |
| $\xi_B$ | number of backcrosses from Ac(WT) to G3 or Ac(DSM)2 |
| $\xi_{GE}$ | binary indicator for genetically engineered strain (Ag(DSM)2 or Ac(DSM)2) |
| $\xi_{WT}$ | binary indicator for wild-type strain Ac(WT) |
| $\xi_J$ | binary indicator for infection by pathogen $j$, for mortality only |

of daily mortality) are bounded (0, 1) and used the complementary log log link function, $h(\psi) = \log(-\log(1 - \psi))$, where $\psi$ is the probability of transmission or daily mortality. On the link-transformed scale, the model used the attributes of table 3 and is given by

$$
\begin{aligned}
\eta = {}& \beta_{G3} + \beta_{WT}\xi_{WT} + \beta_{GE}\xi_{GE} + \beta_G\xi_G + \xi_{GE}(\beta_{GE:G}\xi_G + \beta_B\xi_B + \beta_{G:B}\xi_G\xi_B) \\
& + \xi_J[\beta_{G3:LF} + \beta_{WT:LF}\xi_{WT} + \beta_{GE:LF}\xi_{GE} + \beta_{G:LF}\xi_G \\
& + \xi_{GE:LF}(\beta_{GE:G:LF}\xi_G + \beta_{B:LF}\xi_B + \beta_{G:B:LF}\xi_G\xi_B)].
\end{aligned}
\tag{3.2}
$$

The first three terms of equation (3.2) address the main effects of the three source strains: G3, the local wild type Ac(WT) and Ag(DSM)2. The next two terms address the effect of insectary breeding and backcrosses. The backcrossing term allows for interactions between Ac(WT) and Ac(DSM)2 through different levels of backcrossing and different levels of insectary adaptation by Ac(WT).

In equation (2.1) and table 1, the mortality rate depends on whether the vector is uninfected ($\mu_k$) or infected with pathogen $j$ ($\mu_{jk}$). For mortality, in equation (3.2), the remaining term allows interactions of the previous terms with the infection status for the pathogen $j$. This captures the effect of infection of pathogen $j$ on the mortality rate if $\xi_J = 1$. In the absence of infection, or if experts believed that the pathogen did not substantively alter mortality, then $\xi_J = 0$ and the GLM reduces to the first five terms of equation (3.2). For all parameters other than mortality, $\xi_J = 0$.

To ease comparison, the wild-type and GE lineages were assumed to have identically maintained reproductive cycles in the insectary to facilitate introgression, although other breeding strategies are possible [7]. The number of insectary generations for the wild-type lineage extended from the uncolonized wild-type ($G = 0$) to the 70th generation of Ac(WT) in the laboratory ($G = 70$). The feasible region of the predictors in table 3 is constrained in the following ways. There is no genetically engineered wild-type lineage. For the GE strain, the number of backcrosses cannot exceed the number of generations of Ac(WT) in the insectary. Also, if there are no backcrosses then there is no contribution of the wild-type strain to G3 and so $\xi_G = 0$ in that case.

## 3.2. Elicitation

Each expert only addressed parameters in table 1 that were judged by the expert to be within their domain of expertise. The parameters $\lambda_k$, $\omega_j$, $\kappa_j$, $r_j$, $H$, $v$ and $v_0$ cancelled out of the causal risk ratio and were not elicited. The remaining parameters of table 1 are well studied in entomology and were judged as acceptable elicitation targets except for the daily mortality rate. Rather than daily mortality rate, a common approach is to instead work with the probability of daily survival, e.g. [36,49], where $p = \exp(-\mu)$, or instead $1 - p$, the probability of daily mortality, as was implemented in this study. The

number of experts who could be interviewed for probabilistic elicitations was restricted given time and resource constraints, and obtaining a statistically robust random sample from the diverse set of research domains was impractical for this study. Candidate experts were therefore non-randomly identified from professional contacts on the basis of (i) their scientific expertise in the three mosquito-vectored pathogens in scope, (ii) representation of research institutions from Africa, Europe and North America, and (iii) independence from the research consortium that developed the Ac(DSM)2 transgenic lineage. Sixteen candidate experts were invited to participate of which seven accepted to attend elicitation sessions held in Burkina Faso, the UK and the USA in July 2017. Out of six participating experts able to attend the elicitation sessions, at least one and no more than five contributed assessments for each parameter. Before contributing their judgement, experts were first provided ethics information and the project scope, description and background. Experts were then educated about subjective probability, including cognitive biases such as anchoring and overconfidence. The definitions of quantiles and central credible intervals were introduced before undergoing practice elicitations on unrelated questions with known answers (see [50,51]).

The above model construction (§3.1) induces a multivariate normal distribution for the linear predictor and, for each design point, a transformed Gaussian distribution was elicited for the mean response [52]. An independent conditional means prior approach was applied where the elicited responses contributed by an expert for a given parameter were assumed conditionally independent given the covariate values [53]. For each scenario assessed by the expert (see §S2.a in the R package vignette available in the electronic supplementary material), the 10th, 25th, 50th, 75th and 90th percentiles of the elicited distribution were presented numerically and graphically in addition to the probability density function for the target $\psi$, conditional on the scenario described by each row of $X$ (§3.1). The expert amended the elicited distribution until it satisfactorily approximated their beliefs.

The vector of elicited means $\delta_0$ and diagonal covariance matrix $\Sigma_0$ were obtained for the independent normal distributions, conditional on the elicitation session design matrix $X$ that is square and full rank (electronic supplementary material, §S2.a), on the linear predictor scale so that $\eta \,|\, X \sim N(\delta_0, \Sigma_0)$. Given that $X$ is a full rank square matrix, the elicited location parameter and covariance matrix that modelled the expert's opinion for the unknown coefficients $\beta$ are then given by [52],

$$\delta = (X^\top \Sigma_0^{-1} X)^{-1} X^\top \Sigma_0^{-1} \delta_0 = X^{-1} \delta_0$$

and

$$\Sigma = (X^\top \Sigma_0^{-1} X)^{-1} = X^{-1} \Sigma_0 X^{-\top},$$

with $p(\beta) \sim N(\delta, \Sigma)$ and $\Sigma$ positive definite. The 36 subjective probability assessments and induced priors for the 12 elicited parameters are available in the electronic supplementary material.

## 3.3. Prior and posterior analysis

*A priori*, each expert's contributed probability model was weighted equally. The prior prediction at different scenarios of insectary generations $G$ and backcrosses $B$ was given by the prior Bayesian model average (BMA, [54]), which was an equally weighted mixture of the prior probability models (electronic supplementary material, §S2.b).

The prior assessments for mortality rate were updated using mark release recapture (MRR) data collected in 2013 at a West Africa study site (Bana Village, Burkina Faso [55]) available for a release of wild-type uncolonized female mosquitoes. MRR experiments are a standard technique to assess mortality rates of mosquitoes [56,57]. The experiment collected local mosquitoes from the field as larvae, reared these individuals to adulthood in the insectary and conducted follow-up post-release collections in accordance with guidance for human health and comfort [58]. MRR data typically suggest constant mortality and exponential decay [57]. The MRR process model assumed exponential decay of released uninfected vectors with mortality rate $\mu_k$ as in equation (2.1*b*) (see §S2.c of the R package vignette in the electronic supplementary material for description of the MRR analysis). The observation model for the MRR followed a binomial distribution with the prior for detection probability informed by a review of MRR studies [56]. The MRR data were not presented to the experts for use during the elicitation session (§3.2). The updating of the experts' probabilistic assessments provided a posterior distribution of uninfected vector mortality conditional on the MRR experiment.

## 3.4. Estimation

Markov chain Monte Carlo (MCMC) was used to obtain posterior samples of the uninfected vector mortality rate (§3.1; electronic supplementary material, §S2.c). The MCMC was performed via the R software [59] package rjags [60]. MCMC convergence was assessed by monitoring of trace plots and calculation of the Gelman and Rubin convergence diagnostic [61], which was close to 1 for parameters in all analyses. Given these MRR data, the model evidence (electronic supplementary material, §S2.b) for each expert was obtained by importance sampling [62]. For each parameter, and for each combination of insectary generation $G = 1, \ldots, 70$ and backcross $0 \leq B \leq G$, joint samples from the BMA prior and posterior distributions were obtained for both the released strain with $G$ insectary generations and $B$ backcrosses and the uncolonized wild-type strain with no insectary generations or backcrosses ($G = B = 0$). The BMA mixtures were sampled by first drawing an expert with the appropriate (prior or posterior) model probability then drawing samples from the joint predictive prior distributions (electronic supplementary material, §S2.b). The joint predictive posterior BMA samples for insectary-colonized mortality rates similarly proceeded with composition sampling [63] conditional on the MCMC output for the uninfected vector mortality rate.

Samples of the net logarithmic losses of §2.2 were obtained from the collection of marginal samples independently drawn from the BMA mixtures of each independent parameter (electronic supplementary material, §S2.d). For comparison across the many permutations of insectary generations $G$ and backcrosses $B$, the reporting of the net logarithmic loss was summarized by minimization of its expected absolute error. The absolute error function for an estimate $\chi$ is defined by $|\log_{10}(\mathcal{R}_0^{j(G>0,\,B\geq0)}/\mathcal{R}_0^{j(G=0,B=0)}) - \chi|$ for $\mathcal{R}_0^{jk}$ with analogous expressions for $\mathcal{R}_s^{jk}$ and $\mathcal{R}_{s'}^{jk}$. The median minimizes the expected absolute error loss [28,64]. Medians of the net logarithmic losses and parameters were estimated from the corresponding Monte Carlo samples. The interquartile distance, which is defined as the difference between the 0.75-quantile and the 0.25-quantile, was similarly estimated as a measure of uncertainty for these point estimates (e.g. [65], Appendix B). The reproducible R code and all necessary data for reproducing analyses and figures are provided in electronic supplementary material, §S2.d.

# 4. Results

## 4.1. BMAs for parameters

The elicited prior BMA for the mortality rate of the uninfected and uncolonized wild-type strain, where $G = B = 0$, is shown in figure 1a with the individual assessments. After conditioning on the MRR data, which were available for uninfected and uncolonized wild-type vectors ($G = B = 0$), the resulting posterior density predicted increased mortality rates (figure 1b). The predictions from Experts 1 and 5 received greater posterior probabilities given these specific data. These two experts a priori gave relatively higher weighting to mortality rates above 0.3 compared with the other experts (the green density curves are above the black BMA mixture density curve for high mortality rates in figure 1a).

The mortality parameter was believed by experts to substantively depend on infection status for lymphatic filariasis (§3.1) because ingestion of microfilariae can increase mortality of the vector (e.g. [66]). The mortality rate $\mu_k$ did not change with infection status for malaria or ONNV. For wild-type vectors uninfected with *Wuchereria bancrofti* filaria, the prior medians for mortality rates slightly increased with the number of generations in the insectary from around $\mu_k \approx 0.10$ at $G = 0$ to $\mu_k \approx 0.12$ at $G = 70$ (only very slight gradation in colour in bottom row of top left plot in figure 2). Conditioning on the MRR data resulted in a higher posterior median for the uncolonized wild-type. The mortality rates decreased with the number of generations in the insectary from around $\mu_k \approx 0.36$ at $G = 0$ to $\mu_k \approx 0.27$ at $G = 70$ (slightly lighter shading in leftmost portion of the bottom row of top right plot in figure 2). For wild-type vectors infected with *Wuchereria bancrofti* filaria, the mortality rate increased with the number of insectary generations. A priori median mortality rates increased from $\mu_{lk} \approx 0.18$ to $\mu_{lk} \approx 0.43$ (visible gradient in bottom row of bottom left plot in figure 2). Posterior median estimates increased from $\mu_{lk} \approx 0.34$ to $\mu_{lk} \approx 0.59$ (bottom row of bottom right plot in figure 2).

Compared with a wild-type strain ($B = 0$, no backcrosses) with the same number of insectary generations $G$, higher mortality rates were generally predicted for the GE strain Ac(DSM)2 (at least one backcross, $B > 0$, figure 2). An exception was for a priori median estimates of mortality rates given infection with *W. bancrofti* at high numbers of insectary generations ($G \approx 70$, figure 2, bottom left). In that instance, the wild-type and GE strains had similar median mortality rates. The greatest differences were predicted for the posterior medians of the mortality rates given infection with *W.*

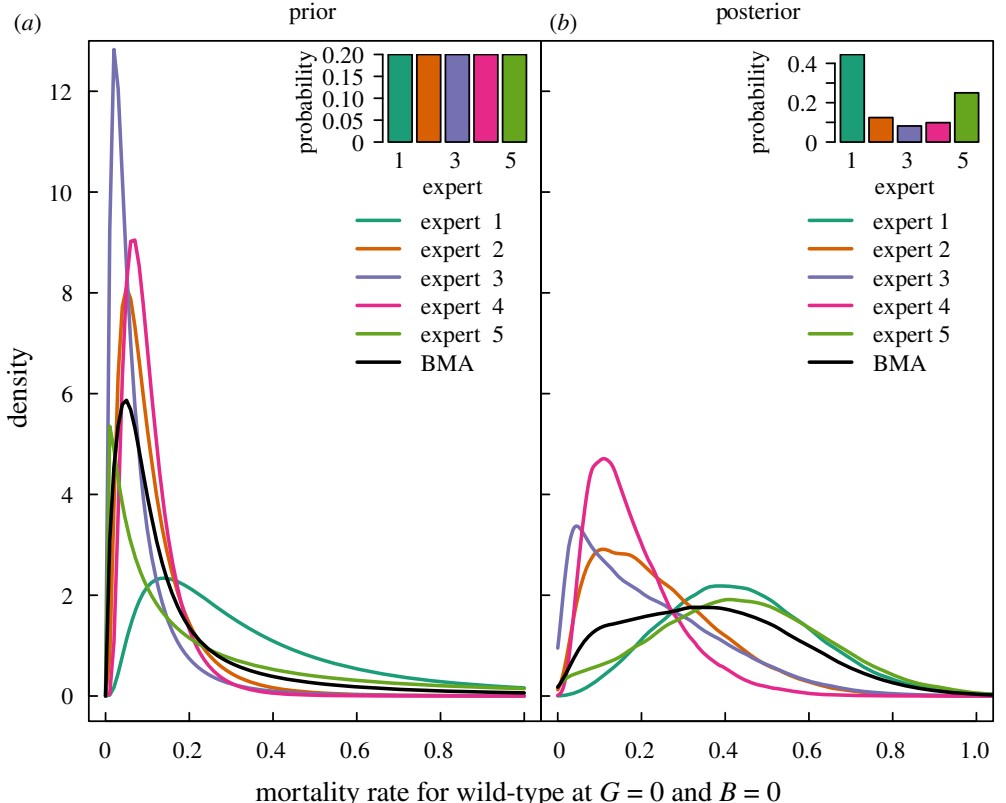

**Figure 1.** Prior and posterior subjective probability densities for the mortality rate $\mu$ of uninfected and uncolonized wild-type vectors with no insectary generations ($G = 0$) or backcrosses with the genetically engineered strain ($B = 0$). Probability density functions are shown by lines and model probabilities by barplots for each expert. The resulting Bayesian model average (BMA) densities are shown by the black lines. All contributing experts were *a priori* weighted equally (*a*). The priors for mortality rate are transformed from the initial probability assessments that targeted the probability of daily mortality $1 - e^{-\mu}$. Conditioning on mark release recapture data increased estimates of mortality rate (*b*).

*bancrofti* (figure 2, bottom right). Estimates for the GE strain had greater uncertainty than wild-type (electronic supplementary material, figure S1).

The predicted median of the human feeding rate decreased with the number of generations in the insectary for the wild-type strain, and was higher for the wild-type than the GE strain (electronic supplementary material, figure S2). The extrinsic incubation periods of lymphatic filariasis and malaria were predicted to be longer for the GE strain compared with the wild-type. For ONNV, the extrinsic incubation period was predicted to be similar for the GE strain and wild-type with a high number of insectary generations, but predictions for uncolonized wild-type with zero or low numbers of insectary generations were accompanied by greater uncertainty (electronic supplementary material, figure S3).

The median transmission efficiencies were generally predicted to be highest for uncolonized wild-type and to decrease with the number of insectary generations (electronic supplementary material, figure S4). Exceptions were the transmission efficiency from vectors to humans for lymphatic filariasis ($b_{lk}$) and ONNV ($b_{ok}$), which were similar for all numbers of insectary generations $G$ for wild-type. The greatest differences in the median estimates were predicted for transmission efficiency from vector to host for malaria ($c_{mk}$), where the wild-type strain had higher predicted transmission efficiency compared with the transgenic strain. The greatest differences in prediction uncertainty between the wild-type and transgenic strain occurred for transmission efficiencies for malaria (electronic supplementary material, figure S5) from both vector to host ($b_{mk}$) and from host to vector ($c_{mk}$). For the former parameter, uncertainty was greatest for transgenic vectors with low numbers of backcrosses and insectary generations. For the latter parameter, uncertainty was greatest for the wild-type strain with no backcrosses.

## 4.2. Net logarithmic loss

Figure 3 describes the results for net logarithmic loss predicted for either choice of index, $\mathcal{R}_0^{(G,B)}$ or $\mathcal{R}_s^{(G,B)}$, as the resulting causal risk ratios are identical (§§2.2 and 2.6). After 70 generations, the net logarithmic

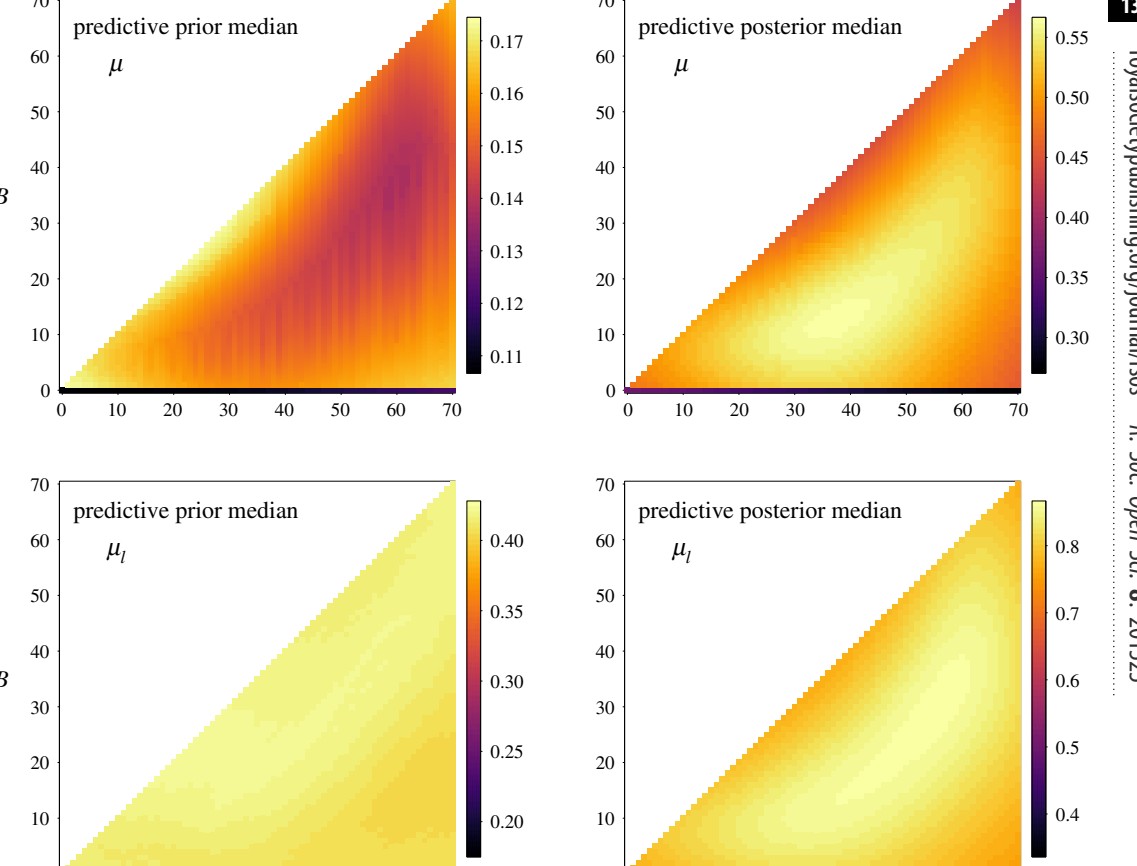

**Figure 2.** Predictive prior and posterior medians of mortality rate (uninfected with filaria, $\mu$, top row; infected with *W. bancrofti* filaria, $\mu_l$, bottom row) for wild-type and genetically engineered strains for different combinations of insectary generations $G$ (x-axes) and backcrosses $B$ (y-axes). Predictive posterior medians (right column) are conditioned on mark release recapture data. In each subfigure, the bottom row corresponds to increasing generations in the insectary $G$ for the wild-type strain with no backcrosses ($B = 0$). The transgenic strain corresponds to at least one backcross ($B > 0$). The top row of subfigures shows that the predicted median mortality rates for uninfected transgenic vectors vary with $G > 0$ and $B > 0$ and are higher than wild-type. The bottom row of subfigures show that the predicted medians increase with insectary generations for wild-type vectors infected with *W. bancrofti*. Note that the number of backcrosses cannot be greater than the number of insectary generations ($B \leq G$).

loss was negative for colonized versus uncolonized wild-type, which implies lower risk of disease transmission for high numbers of insectary generations, with the prior median net logarithmic loss predicted to be less than −0.38 for all three diseases (figure 3). This estimate implies that after 70 generations the number of expected secondary infectious cases decreases by more than half (i.e. $10^{-0.38} = 0.41$) compared with the uncolonized wild-type for all pathogens. For malaria and ONNV, the posterior estimates of the net logarithmic loss for the wild-type strain at 70 generations were slightly higher than the prior estimates, but still less than −0.057, which corresponds to a more than 10% reduction in transmission relative to uncolonized wild-type. For lymphatic filariasis, disease transmission was predicted to decrease by over an order of magnitude compared with uncolonized wild-type after 70 generations for both prior and posterior estimates (figure 3, top row). These results indicate relatively low pathogen transmission for high numbers of insectary generations in the colonized wild-type strain.

The posterior medians of the net logarithmic loss for wild-type infectious for malaria and ONNV increased slightly (corresponding to less than 10% increase in transmission) at small numbers of insectary generations compared with uncolonized wild-type, which may be related to the relatively high posterior estimate of mortality for the uncolonized wild-type compared with the *a priori* predictions (figure 1). The basic reproduction number is sensitive to the mortality rate that appears

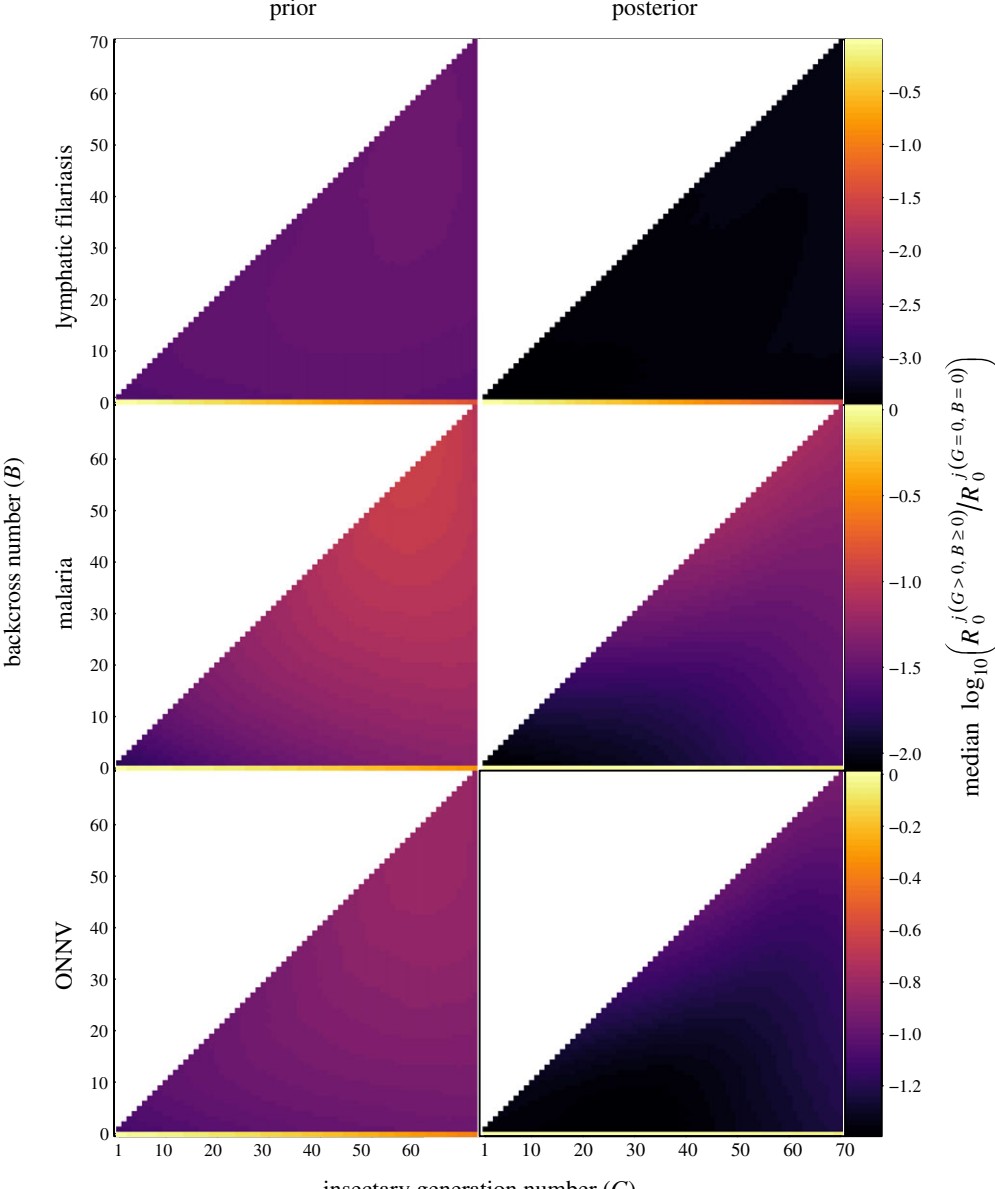

**Figure 3.** Predicted median net logarithmic loss for the $j$th disease (lymphatic filariasis, malaria, ONNV) and the colonized vector strain with at least one insectary generation ($G > 0$; $x$-axis) that is either wild-type with no backcrosses ($B = 0$; $y$-axis) or genetically engineered with at least one backcross ($B > 0$; $y$-axis). Each combination of $G$ and $B$ is compared with an equivalent release of uncolonized wild-type with no insectary generations or backcrosses ($G = B = 0$) to obtain the prior median net logarithmic loss (left). Posterior median net logarithmic loss (right) is conditioned on mark release recapture data that informed vector mortality. The bottom row of each subfigure corresponds to the wild-type strain ($B = 0$); all other rows correspond to the transgenic strain ($B > 0$). For example, the predicted net logarithmic loss is greatest for wild-type ($B = 0$), which has lighter colours (corresponding to greater risk) compared with the transgenic strain with $B > 0$. Prior predictions (left column) for net logarithmic loss decrease as insectary generations $G$ increase for the wild-type strain ($B = 0$). For malaria, increasing the number of backcrosses $B$ is predicted to increase the net logarithmic loss for the transgenic strain (and so decrease the difference in risk compared with uncolonized wild-type). Note that the number of backcrosses cannot be greater than the number of insectary generations ($B \leq G$).

both in an exponential function and also in the denominator of equation (2.2). The posterior medians for the mortality rate of vectors uninfected with filaria decreased with insectary generation (figure 2, top right), which in turn may have influenced the posterior median predictions for net logarithmic loss. Predictions for the mortality rate otherwise increased with insectary generation for the wild-type strain (§4.1).

Overall, the prior median net logarithmic losses suggested that the number of secondary infectious cases for the uncolonized wild-type strain was from 0.8 to 2.6 orders of magnitude greater compared with the transgenic strains (figure 3; electronic supplementary material, table S1). The greatest predicted differences were obtained for lymphatic filariasis (figure 3, top row). Conditioning on the MRR data (figure 3, right) reduced the posterior median net logarithmic loss of the GE strain relative to the prior medians (figure 3, left) for all combinations of backcrosses ($B > 0$) and insectary generations ($G > 0$). The posterior median estimates of the net logarithmic loss ranged between 0.9 to 3.5 orders of magnitude more cases for the uncolonized wild-type compared with the transgenic strain (electronic supplementary material, table S1). For malaria and ONNV, the prior and posterior median net logarithmic loss predicted for transgenic vectors was greater for large numbers of backcrosses (figure 3, right) and hence increased introgression of the local wild-type genetic background into the transgenic strain. The uncertainty associated with the estimated net logarithmic loss as measured by the interquartile distance was always greater for the transgenic strains compared with the wild-type strain with the same number of insectary generations (electronic supplementary material, figure S6).

An alternative comparison is given by the scenario of a release of a single cohort of infectious vectors, described by $\mathcal{R}_s^{j(G,B)}$, compared against a background extant wild population, described by $\mathcal{R}_0^{j(G=0,B=0)}$ (§2.6). For an example release of 10 000 insectary-colonized males in an entomological study, which is twice the number of released uncolonized males in the MRR experiments from the study in [55], a sex separation sorting method with at least 99.5% accuracy [67] results in a maximum of $v_0 = 50$ accidentally released females. At the site of the MRR experiment, the background population of wild-type male mosquitoes is between 10 000 and 50 000 individuals at the end of the dry season increasing to 100 000–500 000 during the wet season [55]. Assuming a sex ratio relatively close to 0.5, the average carrying capacity of local uncolonized wild-type females is probably greater than $v = 5000$. From equation (2.6), the risk ratios $\mathcal{R}_0^{j(G>0,B\geq0)}/\mathcal{R}_0^{j(G=0,B=0)}$ are then reduced by at least a factor of $v_0/v = 0.001$ for the corresponding risk ratio of the single released cohort given by $\mathcal{R}_s^{j(G>0,B\geq0)}/\mathcal{R}_0^{j(G=0,B=0)}$. This cohort comparison leads to a reduction of the net logarithmic losses in figure 3 by a constant $\log_{10}(v_0/v) = -3$, such that the median net logarithmic loss for the single cohort never exceeds $-3$ for the insectary-colonized strains, whether wild-type or transgenic (electronic supplementary material, table S2). If the vectors are assumed uninfected by a pathogen while in the insectary, the net logarithmic loss then further decreases from the bound $\mathcal{R}_s^{j(G>0,B\geq0)}$ exponentially as either the infected vector mortality rate $\mu_{jk}$ or the duration of the extrinsic incubation period $\tau_{jk}$ increases (§2.7).

# 5. Discussion

Scientific risk assessment for a release of transgenic vectors is an interdisciplinary exercise that draws upon the diverse disciplines of decision theory (§2.2), dynamic system modelling (§2), statistical modelling (§3) and the domains of vector-borne disease ecology and genetic engineering. In this analysis, a biomathematical delay differential model was used to formalize the conceptual model scenario and derive measurable risk assessment endpoints. The loss function accounted for ratios of these endpoints as a causal risk ratio that attributes relative risk from exposure to transgenic versus wild-type vectors.

Moreover, the construction of the loss function allowed the expert elicitation to focus on entomological transmission rate parameters (table 1). Human dynamics such as the intrinsic incubation periods and environmentally sensitive variables such as vector larval emergence rate and carrying capacities were cancelled from the elicitation, thus broadening the relevance of expert assessments to the study region in West Africa. Notably, the number of released vectors also cancels out for comparison of transitory disease transmission, which makes the risk assessment results applicable whether the initial release is below or above the carrying capacity: a small release may be of importance for research purposes, for example, whereas a large release might be desired in a genetic vector control setting. The elicitation permitted coherent updating of the expert assessment with field mark release recapture (MRR) field data in a fully Bayesian framework that incorporated both domain knowledge and empirical data. The resulting expected loss estimation therefore supports decision-making that explicitly accounts for uncertainty and risk [27].

Three indices for the relative risk of disease transmission were derived from the mathematical model of equation (2.1). The first is based on the basic reproduction number $\mathcal{R}_0^{(G,B)}$. This index is the long-term average of disease transmission for a vector population release into an empty or vacated niche after $G$

insectary generations and $B$ backcrosses between the wild-type and transgenic strain. The basic reproduction number is perhaps most suitable for assessment of transgenic strains where the genetic construct may persist after release, for example, as might occur for a successful population replacement gene drive system. Relative risk estimates based on $\mathcal{R}_0^{(G,B)}$ are less suitable for other genetic technologies, such as self-limiting SIT, that only persist for a limited number of generations after release.

Therefore, a second index, $\mathcal{R}_s^{(G,B)}$, was derived that focuses on transient disease transmission risk for a single released cohort. The focus on a single release cohort allows consideration of transgenic strains where the genetic construct is expected to quickly decay from a released population, as for a single SIT release. This latter index is relevant, however, for any genetic technology where the immediate short-term relative risk of disease transmission is of interest. The causal risk ratios and net logarithmic loss are shown to be identical for either choice of emphasis on long-term or transitory disease transmission, such that $\mathcal{R}_0^{(G>0,B\geq0)}/\mathcal{R}_0^{(G=0,B=0)} = \mathcal{R}_s^{(G>0,B\geq0)}/\mathcal{R}_s^{(G=0,B=0)}$ (§§2.6 and 4.2). Many of the parameters that determine rates of disease transmission, such as carrying capacity $v$ and the size of release $v_0$, cancel out of these causal risk ratios and so did not require elicitation (table 1).

Importantly, insectary facilities implement safety precautions to avoid infection of insectary vectors by pathogens [7]. A third index $R_{s'}^{(G,B)}$ of transitory disease transmission is proposed to account for these precautions. The indices $R_s^{(G,B)}$ and $R_{s'}^{(G,B)}$ both describe the transient disease transmission risk for a single released cohort. However, the index $R_{s'}^{(G,B)}$ additionally accounts for the uninfected status of a released cohort (§2.7). In the assured absence of infection of vectors by a pathogen in the insectary, the transmission potential indicated by $\mathcal{R}_{s'}$ then exponentially decreases from $\mathcal{R}_s$ with either the vector mortality rate or the duration of the extrinsic incubation period (§2.7). The stringent safety precautions used by insectaries therefore further reduce the potential for transmission by a released cohort to a significant degree.

Many, but not all, of the transmission rate parameters that contribute to the disease transmission indices were predicted by experts to depend on the number of generations in the insectary for the wild-type strain (e.g. [18,19]) and the number of backcrosses with wild-type for the genetically engineered strain. For the wild-type strain, high numbers of generations in the insectary were predicted to decrease transmission. Median estimates of lymphatic filariasis transmission were predicted to decrease by more than an order of magnitude compared with uncolonized wild-type. Estimated disease transmission was typically predicted to be more than an order of magnitude greater for uncolonized wild-type compared with the genetically engineered strain; this difference was further widened by conditioning on female mosquito MRR field data.

The *a priori* predictions for mortality remain relevant as the study scope considered West Africa over both wet and dry seasons, whereas the MRR experiment occurred at a single locale and time. The spatial scope was for all probability assessments defined as the Sudanian zone of Burkina Faso and the neighbouring Sudano-Guinean area of West Africa (§1.2). In this study, the six scientific experts contributed 36 separate probability assessments for 12 elicited parameters derived from the commonly applied Ross–Macdonald modelling framework [34–36] at the defined spatial scope. These assessments are available in the electronic supplementary material to enable full reproducibility of the analyses and also support further research and development of priors for future empirical studies. These priors may be used to support more complex models of vector population dynamics that elaborate specific aspects of the Ross–Macdonald model framework, for example, interspecific density dependence in the *Anopheles gambiae* sensu lato species complex [68]. A more localized study might also incorporate additional information, such as temperature and the local genetic composition of vectors that may improve predictions for parameters such as the extrinsic incubation period [69]. Monitoring data may be incorporated into Bayesian models with environmentally driven parameters, such as time-varying carrying capacity, to assess specific intervention efforts at a location [70]. Integrating empirical information obtained by experiments and monitoring into staged probabilistic risk assessments is discussed in the context of synthetic gene drive in [71].

Current guidance for risk assessments of transgenic vectors recommends evaluation of the relative risk to epidemiologically relevant outcomes such as vectorial capacity by comparison of transgenic versus wild-type vectors [2,21,22]. The causal risk ratio considered by the loss function (§2.2) extends applicability of the risk assessment in space and time, as discussed above, while focusing attention on the entomological parameters of disease transmission. The absolute risk, however, depends on the baseline [41]. Also, parameters such as vector carrying capacity that cancelled out of the causal risk ratios become important when considering absolute risk. Thus, for example, a high relative risk associated with a release may have negligible public health impacts if either the local established

uncolonized wild-type population abundance (described in the model by the carrying capacity $v$) or the size of the released cohorts ($v_0$) is small. The carrying capacity $v$ and size of release $v_0$ also affect the relative risk of a single release of colonized vectors into an empty or vacated niche, described by $\mathcal{R}_s^{(G,B)}$ or $\mathcal{R}_{s'}^{(G,B)}$, compared with the long-term rate of disease transmission for the uncolonized wild-type described by the basic reproduction number $\mathcal{R}_0^{(G=0,B=0)}$ (§4.2). Carefully constructed mark release recapture experiments (e.g. [55,58]), and proxies of abundance obtained by genetic methods [72], may be used to empirically assess vector abundance and carrying capacity at the site of release.

# 6. Conclusion

The framework developed here is applicable to a general assessment of the relative risk of disease transmission by transgenic versus wild-type vectors. The risk assessment supports evidence-based decision-making under uncertainty by evaluation of the losses or utilities among competing alternatives [27,28]. The elicited probabilistic assessments contributed by experts in this study allowed for testable predictions of transgenic vectors in the field. Bayesian model averaging [54] allowed weighting of expert predictions by their ability to predict relevant experimental data, which here informed the mortality rate for the uninfected and uncolonized wild-type strain that was introgressed into the transgenic lineage. A release of transgenic vectors was predicted to produce between a sixfold to over three orders of magnitude reduction in the number of secondary infectious human cases compared with a similar release of the uncolonized wild-type. This risk assessment approach thereby provides a pragmatic step towards a quantitative, evidence-based procedure that estimates the relative harm on human health imposed by transgenic versus unmodified vectors.

Ethics. The expert elicitation was conducted under informed consent from the experts with ethics approval from the CSIRO Social Science Human Research Ethics Committee (application number 065/17). The mark release recapture data provided by Institut de Recherche en Sciences de la Santé, Bobo-Dioulasso, Burkina Faso, was collected with approval from the local institutional ethics committee (Centre Muraz Institutional Ethics Committee), reference number 009–2012/CE-CM.

Data accessibility. The data and code needed to reproduce all analyses and figures are available in the electronic supplementary material as an R package, rRiskGEvec.

Authors' contributions. G.R.H. conceptualized and carried out the original draft preparation, mathematical and statistical analyses; A.I. conducted validation of code; all authors participated in expert elicitation, drafting and revision of the manuscript. All authors gave final approval for publication and agree to be held accountable for the work performed herein.

Competing interests. We declare we have no competing interests.

Funding. This work was supported by the Foundation for the National Institutes of Health and CSIRO Health and Biosecurity.

Acknowledgements. This study would not be possible without the contribution of the independent scientific experts; we thank Yaya Coulibaly (Faculty of Medicine and Odontostomatology, Université des sciences, des techniques et des technologies de Bamako), Roch Dabiré (Institut de Recherche en Sciences de la Santé), Heather Ferguson (Institute of Biodiversity, Animal Health and Comparative Medicine, University of Glasgow), Stephen Higgs (Biosecurity Research Institute, Kansas State University), Steve Lindsay (Department of Biosciences, Durham University) and David Smith (Department of Health Metrics Sciences, School of Medicine, University of Washington) for contributing the subjective probability distributions. This research has made use of mark release capture data provided by the Institut de Recherche en Sciences de la Santé, Bobo-Dioulasso, Burkina Faso. We thank CSIRO Health and Biosecurity, Data61, the Foundation for the National Institutes of Health, and Target Malaria for their help and support. We thank Nicholas Beeton, Dan Gladish and two anonymous reviewers for their helpful comments and suggestions that improved the paper.

Disclaimer. The views expressed in this paper are the authors' own and do not necessarily represent those of the independent scientific experts, funders or acknowledged contributors.

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
