## [Peer Review File · Royal Society Open Science]

Review History

RSOS-201525.R0 (Original submission)

Review form: Reviewer 1

Is the manuscript scientifically sound in its present form?

Yes

Are the interpretations and conclusions justified by the results?

Yes

Is the language acceptable?

Yes

Do you have any ethical concerns with this paper?

No

Have you any concerns about statistical analyses in this paper?

No

Recommendation?

Major revision is needed (please make suggestions in comments)

Comments to the Author(s)

In this manuscript, the authors seek to provide a theoretical framework that can be used to aid in the risk assessment of genetically modified disease vectors - developed here for anopheline mosquitoes. In particular, this study uses mathematical and statistical modelling approaches to assess the pathogen transmission potential of genetically modified vectors relative to their wild-type counterparts. This is achieved through the derivation of basic reproduction number expressions under a range of scenarios and a Bayesian approach to obtain relevant parameter values based on experimental data and expert opinion. Overall I believe that this manuscript seeks to address some important questions for the advancement of this field, however I have outlined a number of concerns below. Based on these I would suggest that this manuscript requires some reasonably heavy revision before it can be published in Royal Society Open Science.

I have worked through most of the calculations within Section 2 (biomathematical model) and been able to derive matching results. I therefore have a lot of confidence in the accuracy of these results. However, I would like to make clear that based on my previous experience I do not feel that I am able to provide a thorough review of the statistical modelling components. As such, I have suggested that the editors seek further review(s) from somebody capable of adequately assessing this section.

Below I have outlined my concerns with the manuscript, split into major and minor points. These are ordered approximately as they appear in the manuscript. Also note that where I state page numbers, I am referring to those in the very top corners (i.e. those listed as Page X of 20).

Major comments:

Data Accessibility statement: The journal website states "It is a condition of publication that authors make the primary data, materials (such as statistical tools, protocols, software) and code publicly available. These must be provided at the point of submission for our Editors and reviewers for peer-review, and then made publicly available at acceptance." However, the mark release recapture (MRR) data used in this study is listed as only available from Patric Epopa "upon reasonable request". There is no indication given as to what restrictions may be placed upon access to this data. The request to Patric Epopa and potential restrictions placed on access to the data also represents a hindrance to the ability of readers to reproduce the results in this manuscript. I also have a serious concern that this data is not listed as being stored in a permanent repository meaning it could be lost to future researchers. Based on the stated aims of the journal I believe this is a serious issue that needs addressing before the article can be published in Royal Society Open Science.

Abstract: Here the abstract reads primarily as a list of results obtained throughout the manuscript. While I appreciate that these are important points, I found it quite hard to interpret what these mean and/or the importance/relevance of these as there is little in the way of motivation given for the study. As I read it I also felt that the methods/approaches used are hinted at but not fully named and/or explained.

Abstract: Here it is stated that "The basic reproduction numbers depend on the number of generations in an insectary colony and the number of backcrosses between the transgenic and wild-type lineages". While I can believe that this statement is likely to be true, so far as I understand it this is a prediction from the model rather than an experimental result and I feel that this should be reflected accordingly. Language of this sort is common throughout the manuscript

and should be re-written to make clear which items are known to be true and what is a model prediction.

Introduction: A number of sections within the introduction feel much more like methods/results sections than introduction sections. I felt that this made for a strange structure to the manuscript. For example, the reader goes through four parts (a through d) of the introduction and then revisit a number of these same concepts in sections 2a-e. To me it felt a bit repetitive and could be improved with some reorganisation.

Page 4, Lines 44-45: I have two comments on the statement around the model scenario being conservative with respect to human exposure to the transgene. (1) I think this statement is incorrect in that humans are not exposed to the transgene, rather they are exposed to vectors carrying the transgene. (2) I'm not 100% sure that the scenario considered is in fact conservative or realistic at all - especially for a sterile release. As I understand it the scenario considered is that where the transgenic individuals completely overtake the wild population and reach the environmental carrying capacity. In reality, what would happen is that transgenic mosquitoes would add to the wild individuals (either via intentional or accidental release). Therefore, the reduced transmission potential of transgenic individuals stated later in this manuscript does not necessarily translate into reduced transmission potential for the overall population (wild and transgenic) as the wild individuals would still be present and likely transmitting at their pre-release level.

Page 8, Line 18: Here is the assumption that the released vector strain becomes established at the release site. For a sterile release this is extremely unlikely to happen - even if they were not fully sterile and the wild-type population were to be eliminated, it is extremely unlikely that the transgenic population could survive in the wild at anywhere close to the environmental carrying capacity - as is assumed here. This should be made clear throughout the study. I would also like to see some consideration of a more realistic scenario - however, I'm aware that this would mean a lot of extra work.

Page 8, Line 40: The assumption that the release does not exceed the environmental carrying capacity seems overly limiting here. For most SIT type approaches it is not uncommon to consider releases that are a multiple of the wild population. This would also be true for certain gene drive approaches. I would say that for this study to be applicable to all of the scenarios the authors have stated in the abstract and introduction, then consideration should also be given to what happens when this restriction is removed. As it stands, this restriction seriously limits the relevance of this study in the real world.

Eqn 6: This equation is fine as stated but only relevant for a very specific (and I believe unrealistic) scenario. This expression compares the R_0 for a transgenic population at environmental carrying capacity and a wild population at the same environmental carrying capacity. As mentioned above, this is unrealistic (especially for release of sterile individuals) as they would only ever add to the wild population (at least where only one or two releases are made) - therefore the true R_0 in the wild would be some combination of the two R_0 values compared here. That being said, I feel that this expression is okay for the splitting of the transgenic and wild-type contributions - but if that is what the author intended then that should be stated explicitly as it is currently not clear.

Page 17, Line 24: This is one example of a feature that occurs a number of times throughout the manuscript. Here the text states "For the wild-type strain, high numbers of generations in the insectary decreased transmission." If I understand it correctly this is a result from the mathematical model - and while seemingly pretty intuitive is not supported here by any experimental data. The language around this point should therefore be adjusted to reflect this

fact. There are a number of occurrences of this type of issue throughout the manuscript and so I would suggest that the authors go through and adjust these accordingly.

Page 17, Line 31: A few points in the manuscript the authors discuss the MRR data from a specific time and place. I think the specific location (either the town/city/village name or coordinates) and a date range for these experiments should be given here to give the reader a full view of what was undertaken.

Conclusions section: The first paragraph of this section seems to be a general discussion on risk assessing transgenic releases and does not actually list any real conclusions so far as I can see. I therefore think this section could be streamlined and/or refocused to draw out more of the key findings of this study.

Minor comments:

Ethics statement: This describes “mark release capture data” but I think it should be “mark release recapture”.

Acknowledgements: The six experts whose opinions were used within this study are stated here. It is not entirely clear whether these were the only experts approached, how they were selected, why they were chosen rather than other experts and how many/which experts were able to contribute opinions for each approximated parameter? I believe that all of these questions need to be answered somewhere in the main text of the manuscript. This would provide full transparency of the work undertaken and I feel is an important addition.

Page 3, Line 21: SIT is usually used to mean sterile insect technique not sterile insect technology as it is stated here.

Page 3, Line 23-24: I agree with the possibility of using SIT for malaria vectors in sub-Saharan Africa. However, since this technology has actually been used successfully in other applications (e.g. New World Screwworm), I feel that these should be mentioned briefly here, just to illustrate the wider potential of the technology.

Page 3, Lines 39-41: Again, I agree with this statement, however it is not just the genetics that may be affected by lab colonisation. For example, changes in the microbiome of insects have been shown to have some interesting effects on a range of behavioural factors. It is up to the authors if they feel this is useful or not but it may be an extra comment to strengthen the argument that lab colonisation can have significant differences on the insects.

Page 4, Line 14: I’m not sure it is enough here to say that the disease free equilibrium is stable. I think the exact nature of this stability (i.e. local/global asymptotic stability) should be what is stated. I believe that further into the manuscript this is stated in more detail.

Page 5, Line 15: It’s a minor concern but it is not ideal to start a sentence with a numerical reference. I wonder whether its worth starting the sentence with something like “Adler [37] provides...”. There is at least one other instance of a reference at the beginning of a sentence and I’d recommend changing that and checking for others also.

Page 5, Line 29: Here $R_0^j(G=0, B=0)$ is listed one way but above it is listed as $R_0^j(0,0)$. I feel that this should be adjusted to be consistent throughout - that is if I’m correct about these representing the same thing.

Page 5, Line 54: “the number of bites on humans per human” sounds a bit clunky as a definition - perhaps something like “the average number of bites on each human”.

Page 5, Lines 56-58: Previously I've seen examples of models where people are assumed to only begin recovery once they have moved into the infected class (rather than during the incubation period). I'm not 100% sure whether that or the authors scenario is correct or whether it is dependent on the pathogen considered. I wonder whether there is any evidence that could be cited for this? This is probably not essential but could be useful for the interested reader.

Table 1: It would be good for the reader and those wanting to reproduce or extend results from this study if this table listed the parameter values that are obtained as the best estimates for each parameter and pathogen.

Page 6, Lines 33-37: I think it would be good here if the assumption that mosquitoes do not survive long enough to recover from infection were to be stated explicitly.

Eqn 2.4a: I think the subscript on r (the one in the exponent) should be j as it is in eqn 2.3a rather than the k it is here.

Page 7, Lines 52-53: This definition is clearly referring to the basic reproduction number however it misses the key point that it is considered in an otherwise completely susceptible population. This point of clarification definitely needs to be added here.

Page 9: The expression at the top of this page appears to assume that a single release results in two transmission generations. I'm not sure I fully understand this assumption because mosquitoes could feasibly survive to blood feed more than once - thereby infecting more than one human. If the authors believe this to be a true assumption, then some further justification and/or relevant citations would be useful here.

Page 13, Line 44: Is there any reason the authors can give as to why the exceptions of transmission efficiency from vectors to humans for lymphatic filariasis and ONNV may occur in reality?

Page 18, Lines 13-14: Here the authors state that the probabilistic assessments contributed by experts allowed testable predictions of what may happen in the field. I have a slight issue here in that this manuscript is supposed to provide a risk assessment framework to be used in advance of releasing transgenic vectors into the field. It seems counterintuitive (to me at least) that one would need to release transgenic vectors into the field in order to test predictions that underpin a risk assessment framework for transgenic releases.

Page 18, Line 15: Is there any more conclusions that can be drawn from the expert predictions. Their initial estimates appear to vary quite significantly which to my mind would suggest one of a few things - e.g. there may not be a good consensus on these processes or they may be lab/location/environment specific. Either of these (or a range of other possible explanations) would suggest that there are further factors that are not captured within this framework. These could possibly be stated as areas for future work. This is not essential, just something for the authors to think about.

Review form: Reviewer 2

Is the manuscript scientifically sound in its present form?

Yes

Are the interpretations and conclusions justified by the results?

No

Is the language acceptable?

Yes

Do you have any ethical concerns with this paper?

No

Have you any concerns about statistical analyses in this paper?

No

Recommendation?

Major revision is needed (please make suggestions in comments)

Comments to the Author(s)

This manuscript describes an approach to evaluate how much transmission would be due to genetically modified vectors in scenarios of control of vector borne diseases.

The manuscript is very well-written.

The methodology is very interesting and sound. Replication code is available, although some data is only available at request.

I think some results can be better explained and demonstrated. Also, the discussion about the significance of the results and how this framework would be applied deserves more attention.

Methodology:

Please explain: "Ignoring the delays in the delayed linear system given by Eq. (2.3)"

R_0 is already stated in Equation 2.2, and presumably derived in Ref. [40], then why all derivation in other equations in section "Threshold Index for Globally Stable Disease Free Equilibrium"? If this is already in the reference I would recommend to cite reference or maybe place this derivation in appendix material.

Equation 2.6 is straightforward given that $v_0 < v$.

Table 3: some parameters are numbers and others are binary indicators. I would prefer different notations for these variables other than using all ϵ s.

Provide more details about the MCMC. How did you check for convergence? How many chains were used and how many iterations.

Results:

Figure 1: credibility intervals for mortality seem very large when looking the posterior distribution. I would recommend describing more uncertainty levels in these results and subsequent results.

Figure 2: Here can you explain better why the triangle shape? If there are for instance 20 generations, then with $B=20$, and $G=20$, there are only backcrosses. This is not clear from observing the plots, when there are no results above the 45 degree line, in this case $G=20$ and $B=20$ (and above).

Also it is difficult to have a good idea of the mortality rate given the gradient of color. I would recommend trying some different visualization. Maybe different plots fixing G and varying B ?

Figure 3 is even more difficult to visualize due to the almost constant level of colors, especially in the posterior. Again, the same comment as above about interpreting the results in varying both G and B values.

Can you present some results about estimation of ϵ variables?

It would be interesting results in appendix about the estimation in MCMC, such as checking for convergence.

Discussion:

Please discuss how abundance estimation, for instance from MRR experiments, can be important for understanding the impact of transmission of genetically modified vectors.

In paragraph discussing the release of 10,000 insectary colonised males in an entomological study, it is said the the risk reduces to $v_0/v = 0.001$. The risk is much reduced, but this means that released number of vectors would be on this order of magnitude. Please discuss some more.

The approach here is parametric in the sense that it is constructed as a function of R_0 and its parameters.

In the field, the number of vectors can fluctuate as well as other conditions, for instance incubation period may vary with temperature. Please comment on the limitations of the present approach given other effects in the field and uncertainty observed in the estimations.

Decision letter (RSOS-201525.R0)

This year has been very difficult for everyone, and we want to take the opportunity to thank you for your continued support in 2020.

The Royal Society Open Science editorial office will be closed from the evening of Friday 18 December 2020 until Monday 4 January 2021. We will not be responding during this time. If you have received a deadline within this time period, please contact us as soon as possible to allow us to extend the deadline. If you receive any automated messages during this time asking you to meet a deadline, we offer apologies and invite you to respond after the festive period or during normal working hours.

With our best for a peaceful festive period and New Year, and we look forward to working with you in 2021.

Dear Dr Hosack

The Editors assigned to your paper RSOS-201525 "Quantifying the risk of vector-borne disease transmission attributable to genetically modified vectors" have now received comments from reviewers and would like you to revise the paper in accordance with the reviewer comments and any comments from the Editors. Please note this decision does not guarantee eventual acceptance.

We invite you to respond to the comments supplied below and revise your manuscript. Below the referees' and Editors' comments (where applicable) we provide additional requirements.

Final acceptance of your manuscript is dependent on these requirements being met. We provide guidance below to help you prepare your revision.

Please submit your revised manuscript and required files (see below) no later than 21 days from today's (ie 15-Dec-2020) date. Note: the ScholarOne system will 'lock' if submission of the revision is attempted 21 or more days after the deadline. If you do not think you will be able to meet this deadline please contact the editorial office immediately.

on behalf of Professor Len Thomas (Associate Editor) and Mark Chaplain (Subject Editor)
openscience@royalsociety.org

Associate Editor Comments to Author (Professor Len Thomas):

Associate Editor: 1

Comments to the Author:

Thank-you for submitting your manuscript to RSOS. We have now received comments from two reviewers and both agree that your work is worthy for publication, but both also have comments and suggestions. I agree, and am therefore recommending that you be given the opportunity to submit a revised version of the manuscript, accounting for these comments. Please include a point-by-point response with your resubmission.

Reviewer 1 noted they did not feel qualified to assess the MRR analysis, so I undertook to do this.

However, there is not enough detail in the main body of the paper to do so, and the Supplementary Materials file, which appears to be written in Sweave or Markdown, references a package `rRiskGVec` that, contrary to what is stated there, does not appear to be available on CRAN either under current or the archive of previously submitted packages.

Reviewer 1 also reports that the data are not available, in contravention with the journal's stated policies.

Therefore, should you choose to resubmit, I ask to you ensure that (1) you document the MRR methods in sufficient detail that they can be reproduced, either in the main paper or Supplementary Materials; (2) that you ensure all code and data are available both for review and

for readers to access after the paper is published. I note that it is not good enough to state that the data will be provided on reasonable request.

I hope you are able to deal in a satisfactory way with these comments, and I look forward to seeing your resubmission.

Reviewer comments to Author:

Reviewer: 1

Comments to the Author(s)

In this manuscript, the authors seek to provide a theoretical framework that can be used to aid in the risk assessment of genetically modified disease vectors - developed here for anopheline mosquitoes. In particular, this study uses mathematical and statistical modelling approaches to assess the pathogen transmission potential of genetically modified vectors relative to their wild-type counterparts. This is achieved through the derivation of basic reproduction number expressions under a range of scenarios and a Bayesian approach to obtain relevant parameter values based on experimental data and expert opinion. Overall I believe that this manuscript seeks to address some important questions for the advancement of this field, however I have outlined a number of concerns below. Based on these I would suggest that this manuscript requires some reasonably heavy revision before it can be published in Royal Society Open Science.

I have worked through most of the calculations within Section 2 (biomathematical model) and been able to derive matching results. I therefore have a lot of confidence in the accuracy of these results. However, I would like to make clear that based on my previous experience I do not feel that I am able to provide a thorough review of the statistical modelling components. As such, I have suggested that the editors seek further review(s) from somebody capable of adequately assessing this section.

Below I have outlined my concerns with the manuscript, split into major and minor points. These are ordered approximately as they appear in the manuscript. Also note that where I state page numbers, I am referring to those in the very top corners (i.e. those listed as Page X of 20).

Major comments:

Data Accessibility statement: The journal website states "It is a condition of publication that authors make the primary data, materials (such as statistical tools, protocols, software) and code publicly available. These must be provided at the point of submission for our Editors and reviewers for peer-review, and then made publicly available at acceptance." However, the mark release recapture (MRR) data used in this study is listed as only available from Patric Epopa "upon reasonable request". There is no indication given as to what restrictions may be placed upon access to this data. The request to Patric Epopa and potential restrictions placed on access to the data also represents a hindrance to the ability of readers to reproduce the results in this manuscript. I also have a serious concern that this data is not listed as being stored in a permanent repository meaning it could be lost to future researchers. Based on the stated aims of the journal I believe this is a serious issue that needs addressing before the article can be published in Royal Society Open Science.

Abstract: Here the abstract reads primarily as a list of results obtained throughout the manuscript. While I appreciate that these are important points, I found it quite hard to interpret what these mean and/or the importance/relevance of these as there is little in the way of

motivation given for the study. As I read it I also felt that the methods/approaches used are hinted at but not fully named and/or explained.

Abstract: Here it is stated that “The basic reproduction numbers depend on the number of generations in an insectary colony and the number of backcrosses between the transgenic and wild-type lineages”. While I can believe that this statement is likely to be true, so far as I understand it this is a prediction from the model rather than an experimental result and I feel that this should be reflected accordingly. Language of this sort is common throughout the manuscript and should be re-written to make clear which items are known to be true and what is a model prediction.

Introduction: A number of sections within the introduction feel much more like methods/results sections than introduction sections. I felt that this made for a strange structure to the manuscript. For example, the reader goes through four parts (a though d) of the introduction and then revisit a number of these same concepts in sections 2a-e. To me it felt a bit repetitive and could be improved with some reorganisation.

Page 4, Lines 44-45: I have two comments on the statement around the model scenario being conservative with respect to human exposure to the transgene. (1) I think this statement is incorrect in that humans are not exposed to the transgene, rather they are exposed to vectors carrying the transgene. (2) I'm not 100% sure that the scenario considered is in fact conservative or realistic at all - especially for a sterile release. As I understand it the scenario considered is that where the transgenic individuals completely overtake the wild population and reach the environmental carrying capacity. In reality, what would happen is that transgenic mosquitoes would add to the wild individuals (either via intentional or accidental release). Therefore, the reduced transmission potential of transgenic individuals stated later in this manuscript does not necessarily translate into reduced transmission potential for the overall population (wild and transgenic) as the wild individuals would still be present and likely transmitting at their pre-release level.

Page 8, Line18: Here is the assumption that the released vector strain becomes established at the release site. For a sterile release this is extremely unlikely to happen - even if they were not fully sterile and the wild-type population were to be eliminated, it is extremely unlikely that the transgenic population could survive in the wild at anywhere close to the environmental carrying capacity - as is assumed here. This should be made clear throughout the study. I would also like to see some consideration of a more realistic scenario - however, I'm aware that this would mean a lot of extra work.

Page 8, Line 40: The assumption that the release does not exceed the environmental carrying capacity seems overly limiting here. For most SIT type approaches it is not uncommon to consider releases that are a multiple of the wild population. This would also be true for certain gene drive approaches. I would say that for this study to be applicable to all of the scenarios the authors have stated in the abstract and introduction, then consideration should also be given to what happens when this restriction is removed. As it stands, this restriction seriously limits the relevance of this study in the real world.

Eqn 6: This equation is fine as stated but only relevant for a very specific (and I believe unrealistic) scenario. This expression compares the R_0 for a transgenic population at environmental carrying capacity and a wild population at the same environmental carrying capacity. As mentioned above, this is unrealistic (especially for release of sterile individuals) as they would only ever add to the wild population (at least where only one or two releases are made) - therefore the true R_0 in the wild would be some combination of the two R_0 values compared here. That being said, I feel that this expression is okay for the splitting of the

transgenic and wild-type contributions - but if that is what the author intended then that should be stated explicitly as it is currently not clear.

Page 17, Line 24: This is one example of a feature that occurs a number of times throughout the manuscript. Here the text states "For the wild-type strain, high numbers of generations in the insectary decreased transmission." If I understand it correctly this is a result from the mathematical model - and while seemingly pretty intuitive is not supported here by any experimental data. The language around this point should therefore be adjusted to reflect this fact. There are a number of occurrences of this type of issue throughout the manuscript and so I would suggest that the authors go through and adjust these accordingly.

Page 17, Line 31: A few points in the manuscript the authors discuss the MRR data from a specific time and place. I think the specific location (either the town/city/village name or coordinates) and a date range for these experiments should be given here to give the reader a full view of what was undertaken.

Conclusions section: The first paragraph of this section seems to be a general discussion on risk assessing transgenic releases and does not actually list any real conclusions so far as I can see. I therefore think this section could be streamlined and/or refocussed to draw out more of the key findings of this study.

Minor comments:

Ethics statement: This describes "mark release capture data" but I think it should be "mark release recapture".

Acknowledgements: The six experts whose opinions were used within this study are stated here. It is not entirely clear whether these were the only experts approached, how they were selected, why they were chosen rather than other experts and how many/which experts were able to contribute opinions for each approximated parameter? I believe that all of these questions need to be answered somewhere in the main text of the manuscript. This would provide full transparency of the work undertaken and I feel is an important addition.

Page 3, Line 21: SIT is usually used to mean sterile insect technique not sterile insect technology as it is stated here.

Page 3, Line 23-24: I agree with the possibility of using SIT for malaria vectors in sub-Saharan Africa. However, since this technology has actually been used successfully in other applications (e.g. New World Screwworm), I feel that these should be mentioned briefly here, just to illustrate the wider potential of the technology.

Page 3, Lines 39-41: Again, I agree with this statement, however it is not just the genetics that may be affected by lab colonisation. For example, changes in the microbiome of insects have been shown to have some interesting effects on a range of behavioural factors. It is up to the authors if they feel this is useful or not but it may be an extra comment to strengthen the argument that lab colonisation can have significant differences on the insects.

Page 4, Line 14: I'm not sure it is enough here to say that the disease free equilibrium is stable. I think the exact nature of this stability (i.e. local/global asymptotic stability) should be what is stated. I believe that further into the manuscript this is stated in more detail.

Page 5, Line 15: It's a minor concern but it is not ideal to start a sentence with a numerical reference. I wonder whether its worth starting the sentence with something like "Adler [37]

provides...". There is at least one other instance of a reference at the beginning of a sentence and I'd recommend changing that and checking for others also.

Page 5, Line 29: Here $R_0^{(G=0, B=0)}$ is listed one way but above it is listed as $R_0^{(0,0)}$. I feel that this should be adjusted to be consistent throughout - that is if I'm correct about these representing the same thing.

Page 5, Line 54: "the number of bites on humans per human" sounds a bit clunky as a definition - perhaps something like "the average number of bites on each human".

Page 5, Lines 56-58: Previously I've seen examples of models where people are assumed to only begin recovery once they have moved into the infected class (rather than during the incubation period). I'm not 100% sure whether that or the authors scenario is correct or whether it is dependent on the pathogen considered. I wonder whether there is any evidence that could be cited for this? This is probably not essential but could be useful for the interested reader.

Table 1: It would be good for the reader and those wanting to reproduce or extend results from this study if this table listed the parameter values that are obtained as the best estimates for each parameter and pathogen.

Page 6, Lines 33-37: I think it would be good here if the assumption that mosquitoes do not survive long enough to recover from infection were to be stated explicitly.

Eqn 2.4a: I think the subscript on r (the one in the exponent) should be j as it is in eqn 2.3a rather than the k it is here.

Page 7, Lines 52-53: This definition is clearly referring to the basic reproduction number however it misses the key point that it is considered in an otherwise completely susceptible population. This point of clarification definitely needs to be added here.

Page 9: The expression at the top of this page appears to assume that a single release results in two transmission generations. I'm not sure I fully understand this assumption because mosquitoes could feasibly survive to blood feed more than once - thereby infecting more than one human. If the authors believe this to be a true assumption, then some further justification and/or relevant citations would be useful here.

Page 13, Line 44: Is there any reason the authors can give as to why the exceptions of transmission efficiency from vectors to humans for lymphatic filariasis and ONNV may occur in reality?

Page 18, Lines 13-14: Here the authors state that the probabilistic assessments contributed by experts allowed testable predictions of what may happen in the field. I have a slight issue here in that this manuscript is supposed to provide a risk assessment framework to be used in advance of releasing transgenic vectors into the field. It seems counterintuitive (to me at least) that one would need to release transgenic vectors into the field in order to test predictions that underpin a risk assessment framework for transgenic releases.

Page 18, Line 15: Is there any more conclusions that can be drawn from the expert predictions. Their initial estimates appear to vary quite significantly which to my mind would suggest one of a few things - e.g. there may not be a good consensus on these processes or they may be lab/location/environment specific. Either of these (or a range of other possible explanations) would suggest that there are further factors that are not captured within this framework. These

could possibly be stated as areas for future work. This is not essential, just something for the authors to think about.

Reviewer: 2

Comments to the Author(s)

This manuscript describes an approach to evaluate how much transmission would be due to genetically modified vectors in scenarios of control of vector borne diseases.

The manuscript is very well-written.

The methodology is very interesting and sound. Replication code is available, although some data is only available at request.

I think some results can be better explained and demonstrated. Also, the discussion about the significance of the results and how this framework would be applied deserves more attention.

Methodology:

Please explain: "Ignoring the delays in the delayed linear system given by Eq. (2.3)"

R_0 is already stated in Equation 2.2, and presumably derived in Ref. [40], then why all derivation in other equations in section "Threshold Index for Globally Stable Disease Free Equilibrium"? If this is already in the reference I would recommend to cite reference or maybe place this derivation in appendix material.

Equation 2.6 is straightforward given that $v_0 < v$.

Table 3: some parameters are numbers and others are binary indicators. I would prefer different notations for these variables other than using all ϵ s.

Provide more details about the MCMC. How did you check for convergence? How many chains were used and how many iterations.

Results:

Figure 1: credibility intervals for mortality seem very large when looking the posterior distribution. I would recommend describing more uncertainty levels in these results and subsequent results.

Figure 2: Here can you explain better why the triangle shape? If there are for instance 20 generations, then with $B=20$, and $G=20$, there are only backcrosses. This is not clear from observing the plots, when there are no results above the 45 degree line, in this case $G=20$ and $B=20$ (and above).

Also it is difficult to have a good idea of the mortality rate given the gradient of color. I would recommend trying some different visualization. Maybe different plots fixing G and varying B ?

Figure 3 is even more difficult to visualize due to the almost constant level of colors, especially in the posterior. Again, the same comment as above about interpreting the results in varying both G and B values.

Can you present some results about estimation of ϵ variables?

It would be interesting results in appendix about the estimation in MCMC, such as checking for convergence.

Discussion:

Please discuss how abundance estimation, for instance from MRR experiments, can be important for understanding the impact of transmission of genetically modified vectors.

In paragraph discussing the release of 10,000 insectary colonised males in an entomological study, it is said the the risk reduces to $v_0/v = 0.001$. The risk is much reduced, but this means that released number of vectors would be on this order of magnitude. Please discuss some more.

The approach here is parametric in the sense that it is constructed as a function of R_0 and its parameters.

In the field, the number of vectors can fluctuate as well as other conditions, for instance incubation period may vary with temperature. Please comment on the limitations of the present approach given other effects in the field and uncertainty observed in the estimations.

===PREPARING YOUR MANUSCRIPT===

===PREPARING YOUR REVISION IN SCHOLARONE===

Author's Response to Decision Letter for (RSOS-201525.R0)

See Appendix A.

Decision letter (RSOS-201525.R1)

Dear Dr Hosack

On behalf of the Editors, we are pleased to inform you that your Manuscript RSOS-201525.R1 "Quantifying the risk of vector-borne disease transmission attributable to genetically modified vectors" has been accepted for publication in Royal Society Open Science subject to minor revision in accordance with the referees' reports. Please find the Editors' comments below my signature.

Please submit your revised manuscript and required files (see below) no later than 7 days from today's (ie 25-Jan-2021) date. Note: the ScholarOne system will 'lock' if submission of the revision is attempted 7 or more days after the deadline. If you do not think you will be able to meet this deadline please contact the editorial office immediately.

Best regards,

on behalf of Professor Len Thomas (Associate Editor) and Mark Chaplain (Subject Editor)
openscience@royalsociety.org

Associate Editor Comments to Author (Professor Len Thomas):

Thank-you for dealing so carefully with the reviewers comments, and for your detailed response. My only remaining requests are related to the rRiskGEvec package submitted as supplemental materials. Please:

1. Export into the namespace all functions required to reproduce the results. In particular this include the `MCMC_jags` function.
2. Provide documentation in the package help for these functions.
3. Make the text of the JAGS model readily available for inspection (currently in lines 63-97 of the file `rRiskGVec_functions.R` in the `.tar` file). One way to do this would be to reproduce the text of the model verbatim in the vignette. Another would be to provide an additional helper function that prints this model. Without this, it requires a fairly advanced R user to find the model code.
4. In line with good programming practice, consider revising these functions so that they do not make use of objects assumed to be in the global workspace, but instead access required objects by passing them in to the functions as function arguments.

===PREPARING YOUR MANUSCRIPT===

===PREPARING YOUR REVISION IN SCHOLARONE===

Attach your point-by-point response to referees and Editors at Step 1 'View and respond to decision letter'. This document should be uploaded in an editable file type (`.doc` or `.docx` are preferred). This is essential.

Author's Response to Decision Letter for (RSOS-201525.R1)

See Appendix B.

Decision letter (RSOS-201525.R2)

Dear Dr Hosack,

It is a pleasure to accept your manuscript entitled "Quantifying the risk of vector-borne disease transmission attributable to genetically modified vectors" in its current form for publication in Royal Society Open Science. The comments of the reviewer(s) who reviewed your manuscript are included at the foot of this letter.

You can expect to receive a proof of your article in the near future. Please contact the editorial office (openscience@royalsociety.org) and the production office (openscience_proofs@royalsociety.org) to let us know if you are likely to be away from e-mail contact – if you are going to be away, please nominate a co-author (if available) to manage the proofing process, and ensure they are copied into your email to the journal.

on behalf of Professor Len Thomas (Associate Editor) and Mark Chaplain (Subject Editor)
openscience@royalsociety.org

Appendix A

Authors' response to comments on original submission: "Quantifying the risk of vector-borne disease transmission attributable to genetically modified vectors"

The authors' point by point responses to comments are provided in italics below.

Associate Editor Comments to Author (Professor Len Thomas):

Associate Editor: 1

Comments to the Author:

Thank-you for submitting your manuscript to RSOS. We have now received comments from two reviewers and both agree that your work is worthy for publication, but both also have comments and suggestions. I agree, and am therefore recommending that you be given the opportunity to submit a revised version of the manuscript, accounting for these comments. Please include a point-by-point response with your resubmission.

Reviewer 1 noted they did not feel qualified to assess the MRR analysis, so I undertook to do this. However, there is not enough detail in the main body of the paper to do so, and the Supplementary Materials file, which appears to be written in Sweave or Markdown, references a package `rRiskGVec` that, contrary to what is stated there, does not appear to be available on CRAN either under current or the archive of previously submitted packages.

Reviewer 1 also reports that the data are not available, in contravention with the journal's stated policies.

Therefore, should you choose to resubmit, I ask to you ensure that (1) you document the MRR methods in sufficient detail that they can be reproduced, either in the main paper or Supplementary Materials; (2) that you ensure all code and data are available both for review and for readers to access after the paper is published. I note that it is not good enough to state that the data will be provided on reasonable request.

I hope you are able to deal in a satisfactory way with these comments, and I look forward to seeing your resubmission.

Authors' response: As part of the request for major revisions, Reviewer 1 recommended restructuring the Introduction, Section 2 and Conclusion. The following structural changes have occurred, which are visible in the tracked changes document:

- *First paragraph of Conclusion section has been moved to start Introduction subsection "Risk Assessment Endpoint"*
- *All paragraphs of the Introduction that introduced mathematical notation have been moved into the first two subsections of Section 2 "Biomathematical Model".*

The main results of predictions for parameters and net logarithmic loss are unchanged from the original submission. However, the reviewers have made many helpful suggestions to improve the presentation of the paper and expand its generality. Our specific responses to each comment from both reviewers are documented point by point below.

The supporting R package with all code and data required to reproduce results and figures is included with the paper as electronic supplementary material. There have been no changes made to the code

in the R package “rRiskGVec”, since there have been no changes to the analyses or figures, but the package documentation has been corrected and revised to improve presentation.

As detailed below, the description of the MRR statistical model in the second paragraph of Section S2.c in the vignette of the R package has been revised. A brief description of the MRR model is also provided here: Consistent with Eq. 2.1b in main text, the MRR process model exhibits deterministic exponential decay of the uncolonised wild-type following release. Given this assumption, the proportion of released vectors that survive each day is equal to $\exp(-\mu_{\{(G=0, B=0)\}})$. The observation model is a binomial distribution with daily sample sizes determined by the unobserved (latent) number of surviving released vectors on each day. The MRR model is hierarchical in construction because there were 3 different batches ($C = 1, 2, 3$) of released vectors to track. In each case, the size of the initial release was known (i.e., sample size of batch C at time $t = 0$ is known). All necessary code and data required to reproduce the MRR model results are included in the R package that accompanies the main document as electronic supplementary material.

The following changes to the R package “rRiskGVec” documentation are listed here:

- Revised vignette to clarify that the package is available as electronic supplementary material to the paper, with all package dependencies available through CRAN
- Data documentation for Epopa et al.’s MRR data (see ?MRR) has been revised to clarify that all MRR data necessary for reproducing all analyses and figures in this paper have been made available in the package. The full MRR data (which includes also male releases not considered by this paper) may be obtained through permission from Patric Epopa.
- “Mark Release Recapture” has been added to the title of section S2.c that describes the MRR data and analysis.
- The following notation substitutions have been updated so that the R package vignette matches the main text (these changes only affect documentation not code):
 - o The notation for the mortality rate in Section S2.c is now explicitly $\mu_{\{(G=0, B=0)\}}$ (previously $\mu_{(0,0)}$), which explicitly denotes zero insectary generations and zero backcrosses because the released female mosquitoes were collected from the field as larvae. See Reviewer 1 comments below recommending that the G, B subscript notation be made explicit throughout. Similar changes have been made in the main text as documented in the point by point responses below.
 - o With the above change of incorporating backcrosses B explicitly into the subscript of the mortality rate in the vignette text of section S2.c, the notation of batch ID number by the variable $B = 1, 2, 3$ required change to avoid conflict. The batch number for released vectors in the MRR experiment is now denoted by C in the vignette.
 - o Corrected typo in second paragraph of Section S2.c: “... the number of surviving females $y(t, C)$ at day [$t > 0$] for each batch ...” (correction from $t \geq 0$ to $t > 0$ is consistent with focus on days $t = 1, 2, \dots$ as specified later in this sentence).
 - o In Section S2.c, the notations for the variables $zeta$ and y have been updated in the revision of the vignette so that these variables correspond to Eqs. 2.1 and 3.1 in the main text: for the MRR analysis, $zeta$ are the daily observed counts of released female mosquitoes as in Eq. 3.1 and y are the actual (i.e., not directly

observed) numbers of released uncolonised wild-type female mosquitoes as in Eq. 2.1.

- *The scenario matrix U in section S2.b of the vignette is now X as in the main text. The vignette text has been adjusted to reflect this notational change by omitting sentences that were specific to the now-omitted matrix U . For completeness, the link function $h(\cdot)$ is also defined in the vignette text (as in Eq. 3.1 of main text).*

The above changes to the vignette are designed to improve its presentation as electronic supplementary material. However, if space constraints in the main text allow it, we note that the material in vignette sections S2.b and S2.c could be moved into the main text. Lastly, while reviewing the above changes for the vignette, out of caution the notation used in the last paragraph of Section 3.b in the main text has also been revised as follows. The elicited mean vector and covariance matrix for linear predictors have been changed to δ_0 and Σ_0 : These parameters were initially defined by m and V , respectively, but the former (m) is also used to denote the vector-host ratio in Ross-Macdonald models of vector-borne disease transmission and the latter (V) is commonly used in the construction of the next generation matrix (FV^{-1}). The revised change of notation to δ_0 and Σ_0 for the elicited mean vector and covariance matrix (of the target parameter ψ transformed to the linear predictor η by the invertible link function h , Eq. 3.1) will help the reader transition from Section 2 (biomathematical modelling) to Section 3 (statistical modelling).

Reviewer comments to Author:

Reviewer: 1

Comments to the Author(s)

In this manuscript, the authors seek to provide a theoretical framework that can be used to aid in the risk assessment of genetically modified disease vectors - developed here for anopheline mosquitoes. In particular, this study uses mathematical and statistical modelling approaches to assess the pathogen transmission potential of genetically modified vectors relative to their wild-type counterparts. This is achieved through the derivation of basic reproduction number expressions under a range of scenarios and a Bayesian approach to obtain relevant parameter values based on experimental data and expert opinion. Overall I believe that this manuscript seeks to address some important questions for the advancement of this field, however I have outlined a number of concerns below. Based on these I would suggest that this manuscript requires some reasonably heavy revision before it can be published in Royal Society Open Science.

I have worked through most of the calculations within Section 2 (biomathematical model) and been able to derive matching results. I therefore have a lot of confidence in the accuracy of these results. However, I would like to make clear that based on my previous experience I do not feel that I am able to provide a thorough review of the statistical modelling components. As such, I have suggested that the editors seek further review(s) from somebody capable of adequately assessing this section. Below I have outlined my concerns with the manuscript, split into major and minor points. These are ordered approximately as they appear in the manuscript. Also note that where I state page numbers, I am referring to those in the very top corners (i.e. those listed as Page X of 20).

Response: We thank the reviewer for the helpful comments and suggested corrections, which have improved the paper. Our responses to each comment are documented below.

Major comments:

Data Accessibility statement: The journal website states "It is a condition of publication that authors make the primary data, materials (such as statistical tools, protocols, software) and code publicly available. These must be provided at the point of submission for our Editors and reviewers for peer-review, and then made publicly available at acceptance." However, the mark release recapture (MRR) data used in this study is listed as only available from Patric Epopa "upon reasonable request". There is no indication given as to what restrictions may be placed upon access to this data. The request to Patric Epopa and potential restrictions placed on access to the data also represents a hindrance to the ability of readers to reproduce the results in this manuscript. I also have a serious concern that this data is not listed as being stored in a permanent repository meaning it could be lost to future researchers. Based on the stated aims of the journal I believe this is a serious issue that needs addressing before the article can be published in Royal Society Open Science.

Response: An R package with all code and data necessary for reproducing analyses was uploaded as supplementary material with our initial submission. All data necessary to reproduce analyses were/are included in the R package with acknowledgments to Patric Epopa and Institut de Recherche en Sciences de la Santé, Bobo-Dioulasso, Burkina Faso and the contributing scientific experts who provided their probability assessments. The statement about the entire MRR dataset, which includes for example male data unused by our study, could be confusing: Its inclusion in the data accessibility statement seemed to suggest that some of the data required for reproducibility are missing (Reviewer 2 and the editor also commented on this). We have therefore moved this section of text from the data accessibility statement into the R package data documentation. The original leading sentence of the statement is however retained in the data accessibility statement so that future researchers know all analyses are full reproducible: "The data, summary statistics and code needed to reproduce these analyses are available in the electronic supplementary material." Our intent is to publish the R package with all code and data necessary to fully reproduce analyses as supplementary material for future researchers. In revising the statement for obtaining the full MRR data, which has been moved to the data documentation of the R package, we have clarified that enquiries for the "full" MRR data should be directed to Patric Epopa, who is the corresponding author for that separate independent study. Those full MRR data include for example male mosquito releases that do not feature in our study.

Abstract: Here the abstract reads primarily as a list of results obtained throughout the manuscript. While I appreciate that these are important points, I found it quite hard to interpret what these mean and/or the importance/relevance of these as there is little in the way of motivation given for the study. As I read it I also felt that the methods/approaches used are hinted at but not fully named and/or explained.

Response: There is a lot more we'd like to say in the Abstract but there is a 200 word limit. We have added "next generation matrix" to the keywords (limited to 6 keywords). In the Abstract, we have added greater emphasis to the general methods contribution by saying that the "... probabilistic risk framework is [demonstrated with] an assessment ..." rather than applied to an assessment.

Abstract: Here it is stated that "The basic reproduction numbers depend on the number of generations in an insectary colony and the number of backcrosses between the transgenic and wild-type lineages". While I can believe that this statement is likely to be true, so far as I understand it this is a prediction from the model rather than an experimental result and I feel that this should be reflected accordingly. Language of this sort is common throughout the manuscript and should be rewritten to make clear which items are known to be true and what is a model prediction.

Response: The abstract word limit is 200 words but we agree that model predictions should be emphasised and have revised here and elsewhere in the text and in the captions of figures 2 and 3.

Introduction: A number of sections within the introduction feel much more like methods/results sections than introduction sections. I felt that this made for a strange structure to the manuscript. For example, the reader goes through four parts (a though d) of the introduction and then revisit a number of these same concepts in sections 2a-e. To me it felt a bit repetitive and could be improved with some reorganisation.

Response: Section 1 was intended to be math-free but still introduced mathematical notation for the key quantities investigated. These expressions and supporting text have been moved out of the Introduction to Section 2, where the mathematical equations are also introduced and explained. The following structural changes have occurred:

- *First paragraph of Conclusion section has been moved to start Introduction subsection "Risk Assessment Endpoint"*
- *All paragraphs of the Introduction that introduced mathematical notation have been moved into the first two subsections of Section 2 "Biomathematical Model".*

Page 4, Lines 44-45: I have two comments on the statement around the model scenario being conservative with respect to human exposure to the transgene. (1) I think this statement is incorrect in that humans are not exposed to the transgene, rather they are exposed to vectors carrying the transgene. (2) I'm not 100% sure that the scenario considered is in fact conservative or realistic at all - especially for a sterile release. As I understand it the scenario considered is that where the transgenic individuals completely overtake the wild population and reach the environmental carrying capacity. In reality, what would happen is that transgenic mosquitoes would add to the wild individuals (either via intentional or accidental release). Therefore, the reduced transmission potential of transgenic individuals stated later in this manuscript does not necessarily translate into reduced transmission potential for the overall population (wild and transgenic) as the wild individuals would still be present and likely transmitting at their pre-release level.

Response: In response to 1), we have clarified "transgene" to exposure to "transgenic vector". For 2), we stress that we are seeking to make a meaningful and fair comparison, and this paragraph is devoted to a thought experiment that allows such a comparison between a release of uncolonized and colonised vectors. The basic reproduction number is about transmission potential in the long term. The two long term scenarios are either wild-type or transgenic vectors persisting at carrying capacity. As we stated, the transgenic vectors are unlikely to persist at carrying capacity due to the fitness cost of the transgene. However, it is not unusual for mathematical models to make restrictive assumptions (for example, constant vector population size, an assumption that we have loosened in this paper) that increase the interpretability of models. We reemphasise the theoretical aspect of R_0 , and omit the word "conservative" to limit comparison with direct field outcomes when using R_0 : "... The transgenic vector would be unlikely to persist at carrying capacity in the long term. The actual exposure to the transgenic vector is likely to be less than what is assumed by an estimate based on the basic reproduction number that assumes a persisting transgenic vector population. Nevertheless, as a theoretical construct \mathcal{R}_0 usefully directs focus to key parameters that determine the transmission potential for vector-borne diseases." We then add a paragraph break before introducing the related indices for a single release, R_s and R_s' , that focus on the transitory effects of a single release of transgenic vectors, which are arguably more fit for purpose.

This latter paragraph now concludes: "The indices $\mathcal{R}_s^{jk}(G, B)$ and $\mathcal{R}_{s'}^{jk}(G, B)$ both focus on the transitory effects of a single released cohort into an empty or vacated niche and hence complement the long term theoretical measure of disease transmission described by the basic reproduction number \mathcal{R}_0 ."

Page 8, Line18: Here is the assumption that the released vector strain becomes established at the release site. For a sterile release this is extremely unlikely to happen - even if they were not fully sterile and the wild-type population were to be eliminated, it is extremely unlikely that the transgenic population could survive in the wild at anywhere close to the environmental carrying capacity - as is assumed here. This should be made clear throughout the study. I would also like to see some consideration of a more realistic scenario - however, I'm aware that this would mean a lot of extra work.

Response: Yes we agree that the long term emphasis of R_0 , although standard procedure for analysis of equilibria in dynamical systems and suitable for establishing transgenic vectors (e.g., self-sustaining gene drive) or insectary colonised wild-type, is not necessarily fit-for-purpose when considering a genetic control release for self-limiting SIT. We made this point earlier in the paper when introducing R_0 but perhaps more emphasis was needed. At the top of Section 2 we've added clarification that underscores the reviewer's point, please see our response above. In the subsection "Evidence Based Decision Making Under Uncertainty", we've added further clarification about the causal risk ratio as a counterfactual measure of effect and added reference to the alternative scenarios of single cohort release (R_s) versus established uncolonised wild-type (R_0) metrics for disease transmission: "The causal risk ratio is an example of a counterfactual or potential outcome definition for effect (Rothman and Greenland 2014, Ch. 3): The causal risk ratio compares disease transmission risk under two different hypothetical scenarios of which one but not the other may actually occur. The basic reproduction number is a long term measure of transmission risk (section 2.e), and so alternative indices of transmission risk are developed for hypothetical scenarios that consider the transitory effect of a released cohort of insectary colonised vectors (section 2.f). These transitory indices and the basic reproduction number are derived from the dynamic system described in the following section."

We also now add further reiteration at the point following the expression where R_0 is defined (Eq. 2.2) : "... However, the transgenic strain is unlikely to persist in the long term, contrary to the assumptions of the basic reproduction number (section 2.a). The next generation matrix and its biological interpretation is therefore applied to develop transitory indices of disease transmission for single cohorts of released vectors, \mathcal{R}_s^{jk} and $\mathcal{R}_{s'}^{jk}$ (see sections 2.f and 2.g) that allow comparison between wild-type and transgenic strains."

At the point in question, we have elaborated again in a new separate paragraph: "The basic reproduction number \mathcal{R}_0^{jk} therefore has the desired interpretation as the average number of secondary infectious hosts for disease j that arise from a single typical infectious host in an otherwise susceptible human population given a released vector strain k that becomes established when introduced into an empty or vacated niche at the release site. However, the transgenic strain is unlikely to persist in the long term (section 2.a). In the following, the next generation matrix is therefore used to derive transitory indices that focus on disease transmission for a single cohort introduced into an empty or vacated niche at the release site."

In the section following, we introduce and interpret the three causal risk ratios considered in the paper (for both long term and short term effects) while referring back to the counterfactual definition for risk provided in the above subsection "Evidence Based Decision Making Under Uncertainty".

Page 8, Line 40: The assumption that the release does not exceed the environmental carrying capacity seems overly limiting here. For most SIT type approaches it is not uncommon to consider releases that are a multiple of the wild population. This would also be true for certain gene drive approaches. I would say that for this study to be applicable to all of the scenarios the authors have stated in the abstract and introduction, then consideration should also be given to what happens when this restriction is removed. As it stands, this restriction seriously limits the relevance of this study in the real world.

Response: This scenario was developed with a research application in mind as described in the Introduction, but we understand the usefulness to generalise the analysis for broader application. We have therefore removed this constraint as suggested. Note that the results (figure 3) are unchanged despite the loosening of assumptions because v and v_0 both cancel out in the relative risk ratios based on either R_0 or instead R_s . In Section "Exposure to Released Cohort", we therefore removed the upper bound for R_0 given by R_s under the restrictive assumption that $v_0 < v$. We added that: "Note that this adjusted bound with v_0 substituted for v in Eq. (2.3b) holds whether or not the number of released vectors v_0 exceeds the carrying capacity v ." Elsewhere in the text we remove mention of R_s as an upper bound on R_0 .

In the beginning of the Discussion, we added extra detail pointing out that the causal risk ratio based on R_0 or instead R_s cancels out the effect of the size of the release, which may exceed carrying capacity in certain genetic control applications, and so generalises the applicability of the risk analysis framework. Also, in the caption of Table 1, we added "The relative risk approach decreased elicitation load and allowed experts to focus on entomological transmission parameters." We also added the size of initial release v_0 to Table 1 and have noted that it did not require elicitation.

Eqn 6: This equation is fine as stated but only relevant for a very specific (and I believe unrealistic) scenario. This expression compares the R_0 for a transgenic population at environmental carrying capacity and a wild population at the same environmental carrying capacity. As mentioned above, this is unrealistic (especially for release of sterile individuals) as they would only ever add to the wild population (at least where only one or two releases are made) - therefore the true R_0 in the wild would be some combination of the two R_0 values compared here. That being said, I feel that this expression is okay for the splitting of the transgenic and wild-type contributions - but if that is what the author intended then that should be stated explicitly as it is currently not clear.

Response: Yes the expression for R_0 is intended to provide an "apples to apples" comparison between uncolonised versus colonised vectors. This use of R_0 as a theoretical means of comparison has been reemphasised at the start of Section 2 as described in the response above. We have removed R_0 from Eq. (2.6), as described above.

Page 17, Line 24: This is one example of a feature that occurs a number of times throughout the manuscript. Here the text states "For the wild-type strain, high numbers of generations in the insectary decreased transmission.". If I understand it correctly this is a result from the mathematical model - and while seemingly pretty intuitive is not supported here by any experimental data. The language around this point should therefore be adjusted to reflect this fact. There are a number of

occurrences of this type of issue throughout the manuscript and so I would suggest that the authors go through and adjust these accordingly.

Response: We note that the topic sentence of this paragraph refers to predictions by experts. We understand the suggestion to reemphasise the predictive aspect of the models in any sentence where results from the models are presented. As a modelling paper, there are many such instances. We now preface presentation with results using words such as "predicted" or "hypothesised" to emphasise the probabilistic and mathematical nature of this study here, in the abstract and in the figure captions.

Page 17, Line 31: A few points in the manuscript the authors discuss the MRR data from a specific time and place. I think the specific location (either the town/city/village name or coordinates) and a date range for these experiments should be given here to give the reader a full view of what was undertaken.

Response: This information was provided in the cited reference, which we now paraphrase in the main text.

Conclusions section: The first paragraph of this section seems to be a general discussion on risk assessing transgenic releases and does not actually list any real conclusions so far as I can see. I therefore think this section could be streamlined and/or refocused to draw out more of the key findings of this study.

Response: The first paragraph of the conclusions has been moved to the beginning of the subsection "Risk Assessment Endpoint" in the Introduction.

Minor comments:

Ethics statement: This describes "mark release capture data" but I think it should be "mark release recapture".

Response: Typo corrected.

Acknowledgements: The six experts whose opinions were used within this study are stated here. It is not entirely clear whether these were the only experts approached, how they were selected, why they were chosen rather than other experts and how many/which experts were able to contribute opinions for each approximated parameter? I believe that all of these questions need to be answered somewhere in the main text of the manuscript. This would provide full transparency of the work undertaken and I feel is an important addition.

Response: We have restructured the introductory paragraph to the subsection "Elicitation" to include information on the expert selection criteria: "... The number of experts who could be interviewed for probabilistic elicitations was restricted given time and resource constraints, and obtaining a statistically robust random sample from the diverse set of research domains was impractical for this study. Candidate experts were therefore non-randomly identified from professional contacts on the basis of 1) their scientific expertise in the three mosquito vectored pathogens in scope, 2) representation of research institutions from Africa, Europe and North America, and 3) independence from the research consortium that developed the Ac(DSM)2 transgenic lineage. 16 candidate experts

were invited to participate of which 7 accepted to attend elicitation sessions held in Burkina Faso, the United Kingdom and the United States in July 2017. Out of six participating experts able to attend the elicitation sessions, at least one and no more than five contributed assessments for each parameter ...” One of the African experts could not attend the elicitation session in Africa due to a cancelled flight and visa restrictions that prohibited rebooking on an alternative flight. The contributing experts are not identified to each parameter as part of the stipulations of ethics approval for the elicitation component of the study. The number of experts for each parameter is available in the accompanying R package, which additionally contains all probability assessments contributed by the experts. There were 36 priors contributed for 12 different parameters, which is too much information to summarise in the main text. An important contribution of the study, however, is that the contributed prior information may also be of use to future researchers (see also comments by Reviewer 2) for insectary colonisation effects and transgenic backcrosses. We therefore explicitly mention in the Discussion that the probability assessments and derived priors for Bayesian GLMs are located within the R package that accompanies the paper as electronic supplementary material: “The six scientific experts contributed 36 separate probability assessments for the 12 elicited parameters. These contributed assessments are available in the Electronic Supplementary Material to enable full reproducibility of the analyses and also support further research and development of priors for future empirical studies.”

Page 3, Line 21: SIT is usually used to mean sterile insect technique not sterile insect technology as it is stated here.

Response: Yes sterile insect technique is the usual term, typo corrected.

Page 3, Line 23-24: I agree with the possibility of using SIT for malaria vectors in sub-Saharan Africa. However, since this technology has actually been used successfully in other applications (e.g. New World Screwworm), I feel that these should be mentioned briefly here, just to illustrate the wider potential of the technology.

Response: We have revised this paragraph to mention that SIT has been used to effectively control insect pests and vectors.

Page 3, Lines 39-41: Again, I agree with this statement, however it is not just the genetics that may be affected by lab colonisation. For example, changes in the microbiome of insects have been shown to have some interesting effects on a range of behavioural factors. It is up to the authors if they feel this is useful or not but it may be an extra comment to strengthen the argument that lab colonisation can have significant differences on the insects.

Response: Yes environmental factors not just genetic factors could affect colonised vectors, although the latter factors are the focus of this study. Environmental by genetic interactions could also occur as discussed in the cited refs. In the Introduction, we have added the recent review by Leftowich et al. 2021, which focuses on both genotypic and phenotypic insectary colonisation effects (including gut microbiota) from the perspective of genetic insect pest management.

Page 4, Line 14: I'm not sure it is enough here to say that the disease free equilibrium is stable. I think the exact nature of this stability (i.e. local/global asymptotic stability) should be what is stated. I believe that further into the manuscript this is stated in more detail.

Response: This text appeared in the Introduction of the initial submission, and refers to R_0 in general terms where it could be applied in settings limited to local (not global) asymptotic stability. This paragraph has been moved to the following section where mathematical details are introduced. Yes in general R_0 refers to local stability. We now introduce "local stability" of the DFE at the top of Section 2 and give it a working interpretation: "... so that disease transmission is not sustained in the long term following the introduction of a small number of infectious individuals." This study does show that R_0 determines global asymptotic stability for this dynamic system.

Page 5, Line 15: It's a minor concern but it is not ideal to start a sentence with a numerical reference. I wonder whether its worth starting the sentence with something like "Adler [37] provides...". There is at least one other instance of a reference at the beginning of a sentence and I'd recommend changing that and checking for others also.

funder

Response: Checked.

Page 5, Line 29: Here $R_0^j(G=0, B=0)$ is listed one way but above it is listed as $R_0^j(0,0)$. I feel that this should be adjusted to be consistent throughout - that is if I'm correct about these representing the same thing.

Response: Yes these represent the same thing. We have maintained the first form as the more informative choice.

Page 5, Line 54: "the number of bites on humans per human" sounds a bit clunky as a definition - perhaps something like "the average number of bites on each human".

Response: If we use "average", we'd then have to define what we're averaging over (e.g., is it a unit of time or area? etc), which could also be clunky. We retain the per capita wording.

Page 5, Lines 56-58: Previously I've seen examples of models where people are assumed to only begin recovery once they have moved into the infected class (rather than during the incubation period). I'm not 100% sure whether that or the authors scenario is correct or whether it is dependent on the pathogen considered. I wonder whether there is any evidence that could be cited for this? This is probably not essential but could be useful for the interested reader.

Response: The dynamics of human recovery from infection is indeed an interesting research question. It is possible that the human recovery rate for some vector borne pathogens could vary during and after the intrinsic incubation period. On the other hand, the construction of the risk ratios has deliberately cancelled this source of uncertainty out of the risk analysis. This cancellation does remain even if the model is generalised to allow for different rates of recovery during and after the intrinsic incubation period. It is straightforward to allow for this possible behaviour of a vector-borne pathogen on human recovery, and so we have done: The new parameter, the human recovery rate during the intrinsic incubation period, is simply multiplied with ω_j in the exponential term of the equation for infectious hosts (Eq. 2.1a) and hence also the indices R_0 , R_s and R_s' . The human recovery rate after the intrinsic incubation period, r_j , is similarly retained in the denominator of these indices (this can be explicitly seen from the next generation matrix FV^{-1} that is fully written out in section 2e, see entry of second row, first column). The risk ratios are unaffected by allowing the

human recovery rate to change at the conclusion of the intrinsic incubation period. Both human recovery rates (within and after the intrinsic incubation period) cancel out from the risk ratios that compare R_0 , R_s and R_s' for colonised versus uncolonised vectors.

Table 1: It would be good for the reader and those wanting to reproduce or extend results from this study if this table listed the parameter values that are obtained as the best estimates for each parameter and pathogen.

Response: The full subjective probability assessments are provided in the R package as supplementary material that will support such future studies. We have added explicit mention of this aspect of the study in Section 3, subsection "Elicitation", and in the Discussion.

Page 6, Lines 33-37: I think it would be good here if the assumption that mosquitoes do not survive long enough to recover from infection were to be stated explicitly.

Response: We have added reference to a susceptible-infectious model in the sentence after the Ross-Macdonald model assumptions for vector borne disease transmission: "The transmission dynamics are modelled by a dynamic system of susceptible and infectious states for both hosts and vectors."

Eqn 2.4a: I think the subscript on r (the one in the exponent) should be j as it is in eqn 2.3a rather than the k it is here.

Response: Yes typo corrected, thank you.

Page 7, Lines 52-53: This definition is clearly referring to the basic reproduction number however it misses the key point that it is considered in an otherwise completely susceptible population. This point of clarification definitely needs to be added here.

Response: We have added the clause "when the host and vector populations are otherwise fully susceptible" to the usual definition for R_0 to account for the vector-borne disease model.

Page 9: The expression at the top of this page appears to assume that a single release results in two transmission generations. I'm not sure I fully understand this assumption because mosquitoes could feasibly survive to blood feed more than once - thereby infecting more than one human. If the authors believe this to be a true assumption, then some further justification and/or relevant citations would be useful here.

Response: The next generation matrix (Diekmann 1990, van den Driessche and Watmough 2002) is defined in terms of transmission generations that allow infection to proceed from hosts to vectors and back. Thus only two transmission generations are required to allow vector-borne disease transmission. A mosquito that becomes infectious may bite multiple humans within the second transmission generation. Above the point in question, in the subsection "Interpretation of Basic Reproduction Number", we have added the next generation matrix FV^{-1} and follow with the explanatory text: "Each entry provides the expected number of infectious secondary cases produced by a primary infectious case in a single transmission generation. For example, an infectious vector is expected to produce $a_{jk} b_{jk} e^{-r_j \omega_j} / \mu_{jk}$ secondary infectious hosts. The expected number of secondary infectious vectors produced by a host while infectious is given by $a_{jk} c_{jk} (v/H) e^{-\lambda \mu_{jk} \tau_{jk}} / r_j$ in a single transmission generation." At the point in

question, we provide reference to the definitions for the next generation matrix and transmission generations in the preceding subsection "Interpretation of Basic Reproduction Number".

Page 13, Line 44: Is there any reason the authors can give as to why the exceptions of transmission efficiency from vectors to humans for lymphatic filariasis and ONNV may occur in reality?

Response: For malaria, median transmission efficiency from host to vector decreased with the number of generations in the insectary for wild-type. This relationship with insectary generation was not predicted for the median for ONNV or lymphatic filariasis. For ONNV, the expert commentary emphasised dependence of transmission efficiency on the viral strain and level of viremia. For lymphatic filariasis, transmission efficiencies were relatively low compared to malaria.

Page 18, Lines 13-14: Here the authors state that the probabilistic assessments contributed by experts allowed testable predictions of what may happen in the field. I have a slight issue here in that this manuscript is supposed to provide a risk assessment framework to be used in advance of releasing transgenic vectors into the field. It seems counterintuitive (to me at least) that one would need to release transgenic vectors into the field in order to test predictions that underpin a risk assessment framework for transgenic releases.

Response: The risk assessment provides testable predictions for a field release. The predictions are quantitative and measurable, so that the models may be assessed scientifically. However, as noted in the introduction and discussion, other models and experiments may be devised for stages prior to release of a transgenic vector, and the release of the transgenic vector may itself be an experiment as part of a larger developmental pathway. Of course these added descriptions are too detailed for a concluding paragraph, but we have restructured the Introduction to highlight that the initial genetic SIT release that was the impetus for this study was part of a larger developmental pathway for gene drive technology to combat malaria. Thus having testable predictions is an important aspect of the risk analysis. In the Discussion we add reference to Hayes et al. (2018), where incorporation of empirical laboratory and field data into staged probabilistic risk assessments are discussed in the context of synthetic genetic vector control strategies. Our main concluding point that we wish to emphasise is simply that the risk assessment predictions developed in this proposed framework may be coherently and empirically assessed.

Page 18, Line 15: Is there any more conclusions that can be drawn from the expert predictions. Their initial estimates appear to vary quite significantly which to my mind would suggest one of a few things - e.g. there may not be a good consensus on these processes or they may be lab/location/environment specific. Either of these (or a range of other possible explanations) would suggest that there are further factors that are not captured within this framework. These could possibly be stated as areas for future work. This is not essential, just something for the authors to think about.

Response: Yes, it is certainly the case that the expert predictions are specific to the context. The expert judgement is conditional on their research experience and the scope of the risk assessment. These assessments should be expected to evolve with the arrival of new data and theoretical developments as would reflect scientific progress. For this study, we provide the assessments as a record of the scientific knowledge assessed at the time, which may hopefully be updated by future studies incorporating new knowledge and information. We added mention of expert assessments made available where the elicitation methods are described and in the Discussion.

Reviewer: 2

Comments to the Author(s)

This manuscript describes an approach to evaluate how much transmission would be due to genetically modified vectors in scenarios of control of vector borne diseases.

The manuscript is very well-written.

The methodology is very interesting and sound. Replication code is available, although some data is only available at request.

I think some results can be better explained and demonstrated. Also, the discussion about the significance of the results and how this framework would be applied deserves more attention.

Response: We appreciate the suggested points for improving the presentation of methods and results, which have improved the paper. Regarding data availability, there appeared to be some confusion caused by our reference of the entire mark release recapture data available from the data owner, IRSS: We reiterate that all of the code and data necessary for reproducing analyses and plotted results are made available in the supplementary R package. The full MRR dataset additionally contains data that are not used by our analyses in this paper, for example, male releases and recaptures. To avoid future confusion, we have moved the data accessibility for the full MRR dataset to the documentation for the female MRR data summaries included in the R package, noting again that all data required for reproducing code are already present in the R package. In the Discussion, we have expanded on the significance of the results by making more frequent direct reference to the risk transmission indices and the results, while also re-emphasising the large spatial scope of the demonstrated risk assessment.

Methodology:

Please explain: "Ignoring the delays in the delayed linear system given by Eq. (2.3)"

Response: We have rephrased: "Setting the delays to zero in the linear system given by Eq. (2.3) ..."

R_0 is already stated in Equation 2.2, and presumably derived in Ref. [40], then why all derivation in other equations in section "Threshold Index for Globally Stable Disease Free Equilibrium"? If this is already in the reference I would recommend to cite reference or maybe place this derivation in appendix material.

Response: Our citation of Ruan et al. (2008), formerly ref [40], did not mean to imply that we have the same model as Ruan et al. Our model differs in that population of vectors may fluctuate in size, following a logistic growth model. Additionally, the equilibrium analysis by [40] does not use the next generation matrix for deriving R_0 . The next generation matrix construction for R_0 is important for our paper because it allows extension to the transitory indices provided by R_s and R_s' . At the top of section 2.d, we have added the following explanatory detail for our approach: "The above expression for \mathcal{R}_0 is similar to the expression derived by (Ruan et al. 2008) generalised here such that the vector population size may vary, the vector mortality rate may depend on infection status, and the human recovery rate may depend on the intrinsic incubation period. In the following, the next generation matrix (Diekmann et al. 1990, van den Driessche and Watmough 2002) is used to derive \mathcal{R}_0 (an alternative derivation is used by Ruan et al. 2008) and thereby inform its biological interpretation for long term disease transmission (section 2.e). The next generation matrix

is then applied to develop transitory indices of disease transmission for single cohorts of released vectors (sections 2.f and 2.g)."

Equation 2.6 is straightforward given that $v_0 < v$.

Response: We have removed this condition in response to the first reviewer.

Table 3: some parameters are numbers and others are binary indicators. I would prefer different notations for these variables other than using all ϵ s.

Response: We understand that the distinction between indicators and the integers is important for interpretation. However, we prefer to emphasise the statistical perspective, where this interpretation does not matter for these covariates as included within the statistical model. Of course, the interpretation of each covariate and its role in the statistical model is indeed important as seen in the following Eq. 3.2.

Provide more details about the MCMC. How did you check for convergence? How many chains were used and how many iterations.

Response: Trace plots and Gelman and Rubin's R statistic were monitored to assess convergence of the MCMC chains. The R statistic diagnostic was calculated in the R code in the supplementary material but not presented in the paper. Yes this is important information that we have added to the main text in the subsection "Estimation".

Results:

Figure 1: credibility intervals for mortality seem very large when looking the posterior distribution. I would recommend describing more uncertainty levels in these results and subsequent results.

Response: In the figure caption we have added emphasis that the mortality rate is plotted rather than the probability of daily mortality $1 - \exp(-\mu)$. We emphasise mortality rate in this figure because it appears in the delay differential equation model. We have added to the caption: "The priors for mortality rate are transformed from the initial probability assessments that targeted the probability of daily mortality $1 - e^{-\mu}$ ". Note also the large spatial scope to which the probability assessments apply. In subsection "Risk assessment endpoint" the spatial scope is defined as the "Sudanian zone of Burkina Faso and the neighbouring Sudano-Guinean area of West Africa", where different environmental conditions and land use may be expected to increase uncertainty in the mortality rate relative to predictions provided for a single point location. We added another sentence describing the specific spatial scope of the probability assessments in the Discussion: "The spatial scope of the probability assessments was defined as the Sudanian zone of Burkina Faso and the neighbouring Sudano-Guinean area of West Africa."

Figure 2: Here can you explain better why the triangle shape? If there are for instance 20 generations, then with $B=20$, and $G=20$, there are only backcrosses. This is not clear from observing the plots, when there are no results above the 45 degree line, in this case $G=20$ and $B=20$ (and above).

Also it is difficult to have a good idea of the mortality rate given the gradient of color. I would recommend trying some different visualization. Maybe different plots fixing G and varying B ?

*Response: We have revised this figure caption and others to provide more guidance to readers on how to interpret these plots, which are complicated given the interactions among strain type, number of insectary generations and number of backcrosses. As suggested by the reviewer, we have also investigated alternative plots for varying insectary generations given fixed backcross, for example, fixing $B = 0$ obtains wild-type for any insectary generation from 0 to 70. However, we think this approach also introduces confusion by proliferating the number of plots for each parameter and also suggesting that the analysis for wild-type was different from the transgenic strain, when in fact both strains are considered together within the same model for a given parameter. After much trial and error, the best solution as we see it is to instead add more explanation to the figure caption that guides the reader to the informative aspects of the plots, so that they do not feel they are missing a possible gradient in colour etc. We have revised the caption to add the following: "In each subfigure, the bottom row corresponds to increasing generations in the insectary G for the wild-type strain with no backcrosses ($B = 0$). The bottom row of subfigures show that the predicted medians increase with insectary generations for wild-type vectors infected with *W. bancrofti*. The transgenic strain corresponds to at least one backcross ($B \geq 0$): The top row of subfigures shows that the predicted median mortality rates are higher compared to wild-type. Note that the number of backcrosses cannot be greater than the number of insectary generations ($B \leq G$)."*

Figure 3 is even more difficult to visualize due to the almost constant level of colors, especially in the posterior. Again, the same comment as above about interpreting the results in varying both G and B values.

Response: Please see above response. We have added the following text to this figure caption: "The bottom row of each subfigure corresponds to the wild-type strain ($B = 0$); all other rows correspond to the transgenic strain ($B > 0$). For example, the predicted net logarithmic loss is greatest for wild-type ($B = 0$), which has lighter colours compared to the transgenic strain with $B > 0$. Prior predictions (left column) for net logarithmic loss decrease as insectary generations G increase for the wild-type strain. For malaria, increasing the number of backcrosses B is predicted to increase the net logarithmic loss for the transgenic strain. Note that the number of backcrosses cannot be greater than the number of insectary generations ($B \leq G$)."

Can you present some results about estimation of ϵ variables?

Response: Rather than present the priors for the unknown coefficients that relate the covariates to the linear predictor of GLMs, which would be difficult to interpret, we instead focus on the more interpretable predictions for the parameters (mortality, transmission efficiency, etc) and how these parameters depends on the number of insectary generations and number of backcrosses. However, as stated all of the prior data were incorporated in the electronic supplementary material, and this information is an important contribution by the scientific experts in this study. In the revision, we have added emphasis that these priors may prove useful for future researchers: There are 36 priors available for GLMs that correspond to 12 different parameters (e.g., mortality rate, transmission efficiencies, etc), which were contributed by the scientific experts. We have added mention of these prior assessments, and their availability in the supplementary R package, in subsection "Elicitation" and in the Discussion.

It would be interesting results in appendix about the estimation in MCMC, such as checking for convergence.

Response: We have added information on convergence of the MCMC to section 3.d "Estimation".

Discussion:

Please discuss how abundance estimation, for instance from MRR experiments, can be important for understanding the impact of transmission of genetically modified vectors.

Response: We mention that the vector host ratio is an important part of R_0 , and also R_s , even though it cancelled out in the relative risk ratio that compares uncolonized wild-type with colonised wild-type and transgenic strains. For absolute risk the vector host ratio becomes again important. MRR is one form of abundance estimation that we have added mention of in the Discussion. We have rewritten the final paragraph before the conclusion to include the following: "Also, parameters such as vector carrying capacity that cancelled out of the causal risk ratios become important when considering absolute risk. Thus, for example, a high relative risk associated with a release may have negligible public health impacts if either the local established uncolonised wild-type population abundance (described in the model by the carrying capacity v) or the size of the released cohorts (v_0) is small. The size of release v_0 and carrying capacity v also affect the relative risk of a single release of colonised vectors into an empty or vacated niche, described by $\mathcal{R}_s^{G, B}$ or $\mathcal{R}_s^{G, B}$, compared to the long term rate of disease transmission for the uncolonised wild-type described by the basic reproduction number $\mathcal{R}_0^{G=0, B=0}$ (section 4.b). Conventional mark release recapture experiments (e.g., Epopa et al. 2017) and proxies of abundance obtained by genetic methods (Jasper et al. 2019), may be used to empirically assess vector abundance at the site of release."

In paragraph discussing the release of 10,000 insectary colonised males in an entomological study, it is said the the risk reduces to $v_0/v = 0.001$. The risk is much reduced, but this means that released number of vectors would be on this order of magnitude. Please discuss some more.

Response: We now reference this comparison in the Discussion, see response above.

The approach here is parametric in the sense that it is constructed as a function of R_0 and its parameters.

In the field, the number of vectors can fluctuate as well as other conditions, for instance incubation period may vary with temperature. Please comment on the limitations of the present approach given other effects in the field and uncertainty observed in the estimations.

Response: We have addressed this point by adding the following to the Discussion: "... The spatial scope was for all probability assessments defined as the Sudanian zone of Burkina Faso and the neighbouring Sudano-Guinean area of West Africa ... In this study, the six scientific experts contributed 36 separate probability assessments for 12 elicited parameters derived from the commonly applied Ross-Macdonald modelling framework (Ross 1911, Macdonal 1957, Reiner et al. 2013) at the defined spatial scope. These assessments are available in the Electronic Supplementary Material to enable full reproducibility of the analyses and also support further research and development of priors for future empirical studies." We note that these prior parameters may help inform the parametrisation of models that are elaborated in other areas, such as interspecific density dependent competition as discussed by Beeton et al. (2020), and also cite a discussion paper (Hayes et al. 2018) on how lab and field data may be incorporated into probabilistic risk assessments for

stagewise development of genetic control strategies. We have also added a reference (White et al. 2011) for using Bayesian inference to estimate environmentally-driven and fluctuating parameters in a process model that may be used to assess specific intervention efforts, and provide a reference for extrinsic incubation period varying with temperature (Ohm et al. 2018).

Appendix B

Authors' response to comments on revised manuscript: "Quantifying the risk of vector-borne disease transmission attributable to genetically modified vectors"

The authors' point by point responses to comments are provided in italics below.

Dear Dr Hosack

On behalf of the Editors, we are pleased to inform you that your Manuscript RSOS-201525.R1 "Quantifying the risk of vector-borne disease transmission attributable to genetically modified vectors" has been accepted for publication in Royal Society Open Science subject to minor revision in accordance with the referees' reports. Please find the Editors' comments below my signature.

Please submit your revised manuscript and required files (see below) no later than 7 days from today's (ie 25-Jan-2021) date. Note: the ScholarOne system will 'lock' if submission of the revision is attempted 7 or more days after the deadline. If you do not think you will be able to meet this deadline please contact the editorial office immediately.

Best regards,

on behalf of Professor Len Thomas (Associate Editor) and Mark Chaplain (Subject Editor)
openscience@royalsociety.org

Associate Editor Comments to Author (Professor Len Thomas):

Thank-you for dealing so carefully with the reviewers comments, and for your detailed response. My only remaining requests are related to the rRiskGExec package submitted as

supplemental materials. Please:

1. Export into the namespace all functions required to reproduce the results. In particular this include the MCMC_jags function.

Authors' response: The supplementary R package in the original submission served to collect data and functions and demonstrate their use in the vignette. It did not include documentation for the functions or the usual package checks. We agree that the suggested revisions will help make the code and data more accessible. All functions are now exported. Additionally, the R package has been revised so that R CMD check succeeds as described in responses below.

2. Provide documentation in the package help for these functions.

Authors' response: All functions are now documented.

3. Make the text of the JAGS model readily available for inspection (currently in lines 63-97 of the file rRiskGVecc_functions.R in the .tar file). One way to do this would be to reproduce the text of the model verbatim in the vignette. Another would be to provide an additional helper function that prints this model. Without this, it requires a fairly advanced R user to find the model code.

Authors' response: The JAGS model in the package has been reimplemented as follows: 1) function is printed in vignette, 2) JAGS model MCMC output, which was previously provided in a compressed file, has been removed from the data subdirectory so that the package can pass RMD check without note, and 3) JAGS model results are now produced when rebuilding vignette. The MCMC run time is ~2 minutes as noted in the vignette. A copy of the R package vignette is included separately from the R package in the supplementary material for those who do not use R and may be unfamiliar with installation of R packages.

4. In line with good programming practice, consider revising these functions so that they do not make use of objects assumed to be in the global workspace, but instead access required objects by passing them in to the functions as function arguments.

Authors' response: The vignette and functions have been revised to avoid use of global variables. All required objects are passed into functions as arguments.